# Integrative Multi-Omics Elucidates the Therapeutic Effect of Coix Seed Oil on Rheumatoid Arthritis via the Gut-Butyrate-Joint Axis and NLRP3 Inflammasome Suppression

**DOI:** 10.3390/ph19010048

**Published:** 2025-12-25

**Authors:** Fanxin Ouyang, Xiaoyu Zhang, Rui Miao, Hongxi Kong, Wenxin Zhang, Zhidan Wang, Xu Han, Shuang Ren, Jie Zhang, Fanyan Meng

**Affiliations:** 1National Regional Diagnosis and Treatment Center for TCM Rheumatology, The First Hospital of China Medical University, China Medical University, Shenyang 110001, China; 2023120925@cmu.edu.cn (F.O.); kong150307@163.com (H.K.); m18842316229@163.com (Z.W.); hanxuzy@126.com (X.H.); renshuang@cmu1h.com (S.R.); 19942057@cmu.edu.cn (J.Z.); 2School of Traditional Chinese Medicine, Liaoning University of Traditional Chinese Medicine, Shenyang 110847, China; shshzlxy@126.com (X.Z.); miaor702@163.com (R.M.); 13841615004@163.com (W.Z.)

**Keywords:** Coix seed oil, rheumatoid arthritis, gut microbiota, butyric acid, NLRP3 inflammasome

## Abstract

**Background:** Rheumatoid arthritis (RA) is a chronic and debilitating autoimmune disease with a complex etiology, creating a significant unmet clinical need for safer and more effective therapeutics. Coix seed oil (CSO), a traditional Chinese medicine with a long history of use against RA, represents a promising candidate; however, its precise mechanisms of action remain largely unexplored. **Objectives:** This study aimed to elucidate the mechanistic basis for the anti-arthritic effects of CSO, with a specific focus on its role in modulating the gut-joint axis. **Methods:** A collagen-induced arthritis (CIA) rat model was employed. The therapeutic efficacy of CSO was evaluated through detailed assessments of arthritic symptoms, joint histopathology, and Micro-CT analysis. To unravel the mechanism, an integrative multi-omics approach was applied, combining untargeted fecal metabolomics with targeted serum metabolomics, which pinpointed butyric acid as a key differential metabolite. This was integrated with 16S rRNA sequencing to profile gut microbiota remodeling. The causal role of butyrate was further verified by exogenous sodium butyrate supplementation in CIA mice. Finally, network pharmacology predictions of potential effector proteins were experimentally validated in vivo using immunofluorescence and qPCR. **Results:** CSO treatment significantly alleviated joint swelling and bone damage in CIA rats after the treatment of 7 days, especially on day 35. CSO primarily restored gut dysbiosis in the CIA model by upregulating butyrate levels, increasing four butyrate-producing probiotics at the genus level, and reducing two pathogenic bacteria. Further exogenous butyrate supplementation validated its ability to improve RA phenotypes. Network pharmacology analysis speculated that there were 142 common targets between CSO and RA, among which NLRP3 was its potential effector protein. In vivo studies verified the suppression of NLRP3 inflammasome activation and reduced expression of subsequent inflammatory mediators by CSO. **Conclusions:** Coix Seed Oil alleviates RA by orchestrating a dual-mechanism action, it remodels the gut microbiota to enhance the production of the microbiotic metabolite butyrate, while also inhibiting the NLRP3 inflammasome pathway. These findings collectively elucidate that CSO mediates its anti-arthritic effects through a novel “gut-butyrate-joint” axis, underscoring its potential as a promising dietary supplement or therapeutic agent derived from medicine-food homology for the management of RA.

## 1. Introduction

Rheumatoid arthritis (RA) is a chronic systemic autoimmune disease defined by persistent synovial inflammation and the progressive destruction of joints and bone [1]. The pathogenesis of RA stems from an overabundance of pro-inflammatory cytokines such as tumor necrosis factor-α (TNF-α), interleukin-1β (IL-1β), and interleukin-6 (IL-6), which contribute to a perpetuating inflammatory microenvironment [2,3]. This chronic inflammation can extend beyond the joints, affecting multiple organ systems, such as the cardiovascular, renal, pulmonary, and nervous systems [4]. Over time, RA can permanently damage joints, lose function, and significantly increase the risk of early death [5,6]. Even if RA can be treated with biotherapy and targeted small molecule medicines, about 40% of people will not fully recover, and some people will slowly deteriorate over time [7,8,9]. In addition, concerns have also been raised regarding the safety of such therapies, especially the increased risk of infection, which remains a major obstacle [10]. Therefore, we urgently need to develop natural, low-toxic, and anti-inflammatory treatments that can not only improve the treatment effect, but also reduce side effects.

In recent years, the emerging concept of the “gut-joint axis” has garnered significant attention, which illustrates the reciprocal relationship governing gut microbiota-immune system interactions [11]. Through this approach, the microbiota in the gut affects joint inflammation through immune and metabolic interactions, an idea similar to the concept of traditional Chinese medicine (TCM) such as “spleen deficiency and dampness obstructing the flow of Qi to the joints” [12]. Dysbiosis, defined as an imbalance in the gut microbiota, not only contributes to the initial onset of RA but also perpetuates a state of persistent systemic inflammation—a hallmark feature of the disease [13,14,15]. In addition, short-chain fatty acids (SCFAs) are key metabolites derived from the microbial fermentation of dietary fiber in the gut. Their growing importance in RA pathogenesis is attributed to their pleiotropic immunomodulatory functions [16]. A growing body of evidence indicates that SCFAs are involved in the pathogenesis of RA, thereby implying their potential role in regulating inflammatory responses [17,18]. The protective role of SCFAs, particularly butyrate, in RA is mediated through multiple interconnected mechanisms. Firstly, they directly recalibrate the immune balance by expanding Treg cells and inhibiting Th17 differentiation [19]. Secondly, they suppress the synthesis of pivotal cytokines like TNF-α and IL-6 [20]. Lastly, by fortifying the gut barrier, they prevent the systemic spread of inflammation [21]. Notably, the observed decrease in SCFAs and SCFA-producing bacteria in RA patients confirms the pathophysiological relevance of these mechanisms [22,23]. Hence, targeting gut microbiota regulation is being considered as a promising strategy for symptom relief and disease management in RA.

RA is traditionally classified as an “arthralgia syndrome” in TCM theory. Current research has found that TCM has a great effect on the core pathological processes that affect RA, which provides us with new treatment ideas [24]. Studies have indicated that treatment with traditional Chinese herbs can help remodel the ecological architecture of the gut microbiome, particularly promoting the proliferation of beneficial bacteria such as *Faecalibacterium*, which performs an indispensable function in the synthesis of SCFAs [25]. Among the SCFAs, butyrate has emerged as a key metabolite, exerting direct inhibitory effects on the NLRP3 inflammasome—an essential mediator of synovial inflammation and joint degradation in RA [26,27,28]. The NLRP3 inflammasome not only contributes to joint inflammation but also exacerbates bone damage, making it a pivotal target for RA therapy [29,30]. Given these insights, TCM-based interventions that target the “gut-joint axis” have garnered increasing attention as promising strategies for RA management, offering novel avenues for paving the way for superior treatment paradigms.

Coix Seed Oil (CSO), obtained through the extraction of the seeds Coix lacryma-jobi L., is a well-known traditional herbal remedy with both medicinal and nutritional value. It has been used for treating inflammatory diseases and rheumatic disorders [31]. The therapeutic potential of CSO is primarily attributed to its unique composition of bioactive lipids. It is notably rich in unsaturated fatty acids, such as oleic acid, linoleic acid, cholic acid and so on, which have demonstrated anti-inflammatory and microbiota-modulatory properties [32,33]. Furthermore, CSO contains specific functional esters, which have been reported to contribute to its spectrum of beneficial bioactivities, such as anti-tumor and immune-regulating capacities [34]. In our previous work, we confirmed the potential of CSO by functioning as a regulator of gut microbiota dysbiosis and an inhibitor of pro-inflammatory cytokine secretion, suggesting its therapeutic value in the management of RA [35,36]. Nevertheless, the specific processes by which CSO modulates gut microbiota composition, increases SCFAs levels, and suppresses inflammatory responses to regulate the “gut-joint axis” in alleviating RA remain unclear.

Therefore, this research explores the investigational effects of CSO in a collagen-induced arthritis (CIA) model, with the goal of elucidating its mechanism in regulating the intestinal microbial community, regulation of SCFAs, and inhibition of the NLRP3 inflammasome pathway. This research seeks to enhance our understanding of the “gut-joint axis” in the pathological processes associated with RA, understand how CSO treats diseases, and provide a theoretical basis for using traditional Chinese medicine to treat RA, with the focus on studying changes in butyric acid within intestinal flora.

## 2. Results

### 2.1. UHPLC-OE/MS Analysis of CSO

A comprehensive characterization of the chemical constituents of CSO was performed using UHPLC-OE/MS. In total, 432 active components were identified. Representative total ion current (TIC) chromatograms of CSO in both positive and negative ionization modes are provided in Appendix A. To highlight the most abundant constituents, the top 10 compounds ranked by peak area in each ionization mode are listed in Table 1 (positive mode) and Table 2 (negative mode). These tables provide detailed information, including composite scores, chemical formulas, retention times, and precise m/z values for these key components.

### 2.2. CSO Alleviated Arthritis Symptoms in CIA Rats

AI and paw thickness in rats directly evaluate the onset and progression of arthritis. As illustrated in Figure 1B,C, both the paw swelling and AI increased significantly in the CIA group by day 7 post-immunization. In contrast, treatment with CSO-M and CSO-H led to a marked reduction in both paw swelling and AI starting from the 14th day (*p* < 0.05). Furthermore, as shown in Figure 1D,E, both the CSO and MTX groups exhibited significant improvements in paw swelling and AI compared to the CIA group on day 35 (*p* < 0.05). These findings suggest that CSO significantly alleviates arthritis symptoms in CIA rats.

There is accumulating evidence linking the overexpression of inflammatory cytokines cascade to the progression of inflammation and imbalance in bone remodeling characteristic of RA [37]. First we evaluated the effect of CSO on inflammatory cytokines by analyzing the levels of inflammatory factors in rat serum using ELISA technology. As shown in Figure 1I–K, the expression of TNF-α, IL-6, and IL-1β was significantly higher in the CIA group compared to the CON group (*p* < 0.01). In contrast, these factors were significantly reduced in the CSO and MTX groups (*p* < 0.05). Furthermore, the expression of TNF -α and IL-6 in the ankle joint tissues was assessed. As shown in Figure 1F–H, their expression was markedly elevated in CIA rats, and CSO significantly down-regulated their levels in a dose-dependent manner (*p* < 0.01). Collectively, these results confirm that CSO significantly alleviated joint inflammation and bone destruction in CIA rats.

Joint pathology and cartilage destruction in CIA rats were evaluated using HE, SOFG, and TRAP staining, respectively. As shown in Figure 2A, the CIA group exhibited marked synovial hyperplasia, an increased number of synovial cell layers, disorganized arrangement, and inflammatory cell infiltration. Treatment with CSO-H and MTX significantly reduced these changes, resulting in lower HE score (Figure 2H, *p* < 0.01). SOFG staining revealed intact articular cartilage in the CON group, with clear cartilage-bone boundaries and no signs of destruction. In contrast, the CIA group showed severe cartilage damage, with scattered chondrocytes and blurred cartilage-bone boundaries (Figure 2B). Quantification of SOFG staining demonstrated that cartilage destruction was significantly reduced in the CSO-H and MTX groups (Figure 2I, *p* < 0.01). Figure 2C indicates that the CIA group displayed numerous osteoclasts, whereas treatment with CSO and MTX reduced osteoclast numbers, as shown by TRAP staining (Figure 2J, *p* < 0.05). These findings confirm that CSO significantly alleviates joint pathology and cartilage destruction in CIA rats.

To further evaluate bone erosion, we employed micro-CT imaging. As shown in Figure 2D, no bone destruction was observed in the paw structure of rats in the CON group. In contrast, the CIA group exhibited severe bone erosion, with honeycomb-like lesions on the ankles and toes, accompanied by joint deformation. Compared to the CIA group, the CSO-M, CSO-H, and MTX groups showed improved joint erosion. Key bone morphology parameters, including BV, BV/TV, and BS/TV, were analyzed in Figure 2E–G. These parameters decreased in the CIA group, while in the CSO and MTX groups, BV, BV/TV and BS/TV were significantly increased (*p* < 0.05). The results show that CSO improved bone erosion.

### 2.3. Abnormal Butyric Acid Metabolism in the Feces and Serum of CIA Rats

To further investigate the mechanisms underlying the development of joint inflammation and bone destruction in CIA rats, we conducted additional analyses of relevant metabolites in their feces. Firstly, we used Orthogonal Partial Least Squares Discriminant Analysis (OPLS-DA) to represent similarities or differences among samples visually. The OPLS-DA model indicated a clear separation between the metabolic profiles of the CON and CIA groups, pointing to marked biochemical alterations induced by CIA (Figure 3A). Then we calculated the quantitative changes of metabolites between the CON group and the CIA group in rat feces through statistical tests (Figure 3B). The number of differentially expressed metabolites in feces was 254 (Fold Change > 2 or < 0.5, *p* < 0.05), and the up-regulated and down-regulated metabolites were 138 and 116, respectively. The top 50 differential metabolites were visualized in a heatmap (Figure 3C). Following this visualization, pathway enrichment analysis for these metabolites was performed based on the KEGG database. KEGG revealed that fatty acid degradation and fatty acid biosynthesis associated with lipid metabolism were significantly differentially expressed (Figure 3D).

To further elucidate fatty acid differences, we performed targeted metabolomics analysis of plasma fatty acids. As illustrated in Figure 3E, subsequent measurement of plasma SCFAs showed that the CIA group exhibited a significant reduction in butyric acid (BA) concentration relative to the CON group (*p* < 0.01). Conversely, acetic acid level was significantly increased in CIA group (*p* < 0.01). In conclusion, SCFAs metabolism is abnormal in the feces of arthritis model rats, particularly with reduced BA levels.

### 2.4. Dysregulated Metabolism of Gut Microbiota with a Reduction in Butyrate-Producing Bacteria in CIA Rats

Butyrate is a short-chain fatty acid produced by gut microbiota metabolism. To further investigate the origins of fecal changes and serum metabolic abnormalities in arthritis models, we conducted an in-depth analysis of the fecal microbiota in CIA rats. This analysis revealed a state of dysbiosis, characterized by a significant reduction in populations of butyrate-producing bacteria. As shown in Figure 4A–D, compared to the CON group, both the CIA group exhibited significant alterations in the α-diversity and β-diversity of the microbiota. Then we studied the quantitative differences in intestinal flora at different classification levels among these groups. The family level (Figure 4E,F) included an enrichment of *Akkermansiaceae* and *Bacteroidaceae*, alongside a depletion of *Lactobacillaceae* and *Peptosreptococcaceae* in the CIA group (*p* < 0.05). At the genus level (Figure 4G,H), *Akkermansia* and *Bacteroides* were enriched in the CIA group, while butyrate-producing *Lactobacillus* and *Limosilactobacillus* were depleted (*p* < 0.05).

### 2.5. CSO Increased the Levels of Butyrate in CIA Rat Serum and the Abundance of Butyrate-Producing Bacteria in the Fecal Microbiota

To investigate whether CSO improves RA by restoring the dysregulated gut microbiota and associated metabolic changes, we conducted analyses of fecal and serum samples from rats following CSO intervention. 16S rRNA gene sequencing of fecal samples from the rat models was performed to assess the impact of CSO on the gut microbiota composition. Compared to the CIA group, both the CSO and MTX groups exhibited significant alterations in the Chao1, Simpson, and Shannon indices (Figure 5A,C,E), indicating that CSO modulates the α-diversity of the microbiota. To evaluate similarities and differences in intestinal microbiota structures across the three experimental groups, we conducted principal coordinate analysis (PCoA) based on UniFrac distances using operational taxonomic units (OTUs). This analysis showed distinct separation of microbiota among the three groups, suggesting that arthritis induces changes in intestinal β-diversity (Figure 5B). As shown in Figure 5D,F, LEfSe analysis identified 46 significantly discriminative genera (LDA score > 3.0).

Then we studied the quantitative differences in intestinal flora at different classification levels among three experimental groups. A significantly lower *Firmicutes/Bacteroidota* (F/B) ratio was observed in the CIA model group at the phylum level compared to controls (Figure 6A,E). This ratio was restored upon intervention with CSO or MTX. Furthermore, taxonomic shifts at the family level (Figure 6B) included an enrichment of *Akkermansiaceae* and *Bacteroidaceae*, alongside a depletion of *Lactobacillaceae* in the CIA group (*p* < 0.05). Both CSO and MTX interventions reversed these trends (*p* < 0.05). At the genus level (Figure 6D), *Akkermansia*, *Bacteroides*, and *Incertae Sedis* were enriched in the CIA group, while butyrate-producing *Lactobacillus*, *Limosilactobacillus*, and *Romboutsia* were depleted (*p* < 0.05). Following CSO and MTX treatment, these patterns were similarly reversed (*p* < 0.05). Species-level composition analysis (Figure 6C) further confirmed these findings, demonstrating consistency at a finer taxonomic resolution (*p* < 0.05). Additionally, KEGG pathway analysis after CSO treatment revealed significant enrichment of RA-associated inflammatory pathways, including fatty acid degradation (Figure 6F). It is worth noticing that ELISA-based quantification of serum butyric acid revealed significantly lower levels in the CIA group compared to the CON group. After CSO intervention, butyric acid levels were restored (Figure 6G). These results collectively indicate reduced butyrate-producing bacteria in CIA, with increased abundance of such microbiota following administration.

To further validate CSO’s improvement of RA via the microbiota-butyrate-joint axis, we employed Spearman correlation analysis. Correlation analysis in Figure 6I similarly demonstrated a positive correlation between butyrate levels and beneficial butyrate-producing bacteria *Lactobacillus.* Concurrently, as shown in Figure 6J (*p* < 0.05), it demonstrated a negative correlation between butyrate levels and inflammatory factors (TNF-α, IL-1β, and IL-6) and pathological scores (HE, SOFG, and osteoclast counts). Furthermore, the relationship between gut microbiota and arthritis symptoms is depicted in Figure 6H (*p* < 0.05). *Lactobacillus*, probiotics producing butyrate, were negatively correlated with the severity of inflammatory factors and pathological scores. Conversely, harmful bacteria such as *Bacteroides* and *Akkermansia* were positively correlated with AI severity scores, foot swelling, inflammatory cytokines and pathological scores. These findings suggest that improvement in gut dysbiosis is a key mechanism underlying the anti-rheumatoid arthritis effects of CSO, potentially achieved through increased butyrate concentration regulation by the microbiota.

### 2.6. Exogenous BA Can Mimic the Effect of CSO, Alleviating the RA Phenotype in CIA Mice

Multi-omics results suggest that butyric acid (BA) plays a crucial role as a mediator in CSO’ beneficial effects on RA. However, its direct role in improving RA still requires further validation. Therefore, we next evaluated the effects of exogenous butyric acid supplementation in the CIA model (Figure 7A). As shown in Figure 7B, both CSO and BA treatments significantly improved body weight compared to the CIA group, accompanied by a marked reduction in paw swelling and AI (Figure 7C,D). Furthermore, as depicted in Figure 7E–G, both the CSO and BA groups demonstrated substantial improvements in body weight, paw swelling, and AI compared to the CIA group (*p* < 0.01). Histological analysis revealed significant foot swelling, synovial hyperplasia, disorganized tissue architecture, and inflammatory cell infiltration in the CIA group. However, treatment with CSO and BA significantly alleviated these pathological changes (Figure 7H,I, *p* < 0.01). Additionally, as shown in Figure 7J,K, both CSO and BA treatments notably reduced the expression levels of the pro-inflammatory cytokines TNF-α and IL-6 (*p* < 0.01). Taken together, our data provide strong evidence that butyric acid effectively alleviates RA symptoms, with therapeutic effects comparable to those of CSO.

### 2.7. CSO Inhibited NLRP3 Activation and Improving Joint Inflammation in CIA Rats

To further explore its potential mechanisms, network pharmacology was employed to analyze the possible effector proteins of CSO in RA via TCMSP, Gene Cards, Swiss Target Prediction, OMIM, and so on. The subsequent analysis revealed a set of 142 core targets connecting CSO to the pathology of RA (Figure 8A). The PPI network of these targets is depicted in Figure 8B. KEGG pathway analysis revealed pathways such as the NOD-like receptor signaling pathway and apoptosis, both of which are regulated by NLRP3 activation in inflammatory responses (Figure 8C). GO enrichment analysis indicated biological processes primarily related to the regulation of inflammatory responses and defense against bacterial infections (Figure 8E). Molecular docking studies demonstrated that three natural compounds from CSO—Coixenolide, Sitosterol, and Stigmasterol—interacted with the NLRP3 protein, with binding energies of −6.401 kcal/mol, −8.896 kcal/mol, and −9.204 kcal/mol respectively (Figure 8D). These findings suggest that the therapeutic effect of CSO on RA may be closely linked to the modulation of the inflammatory response through NLRP3 regulation.

To further validate the regulatory mechanism of NLRP3 inflammasome following medicine administration by detecting the expression of NLRP3 and its downstream proteins in mice ankle joint tissues via RT-qPCR (Figure 9E–J). Quantitative analysis revealed that the expression levels of NLRP3, Caspase-1, GSMDM, IL-1β, IL-18, and ASC in the CIA group were significantly higher than those in the CON group (*p* < 0.01). After CSO treatment, a gradient decrease in the expression of NLRP3 activation-related proteins was observed (*p* < 0.05). Moreover, to further evaluate the regulatory impact of CSO on NLRP3 inflammasome activation, we assessed the expression of NLRP3, Caspase-1 and IL-18 in ankle joint sections from each group using immunofluorescence staining. As illustrated in Figure 9A–D, the fluorescence intensity of NLRP3, Caspase-1, and IL-18 in the CIA group was significantly higher than that in the CON group, indicating marked over-expression of these proteins in the CIA group. Furthermore, the fluorescence signals of the associated inflammatory factors showed a notable reduction following intervention with medium and high doses of CSO, as well as MTX, thereby reinforcing the reliability of our findings. It is worth noting that through correlation analysis (Figure 9K), it was found that the beneficial bacteria that produce butyric acid and the serum butyric acid level were negatively correlated with the expression of NLRP3 and its downstream factors, and the harmful bacteria were negatively correlated with it. Meanwhile, in mice, the qPCR results of CSO improving NLRP3 were inversely proportional to the improved serum butyric acid level. In conclusion, CSO can effectively inhibit the activation of the NLRP3 inflammasome.

## 3. Discussion

In recent years, bioactive components from food-medicine dual use sources have demonstrated considerable potential in the treatment of inflammatory diseases, including RA [38]. To elucidate the material basis for the therapeutic effects of CSO, we employed UHPLC-OE/MS analysis to characterize its chemical composition. The results revealed that CSO is rich in several bioactive compounds, predominantly Indirubin, Oleic acid, and Linoleic acid. Notably, these compounds have been extensively reported to possess potent anti-inflammatory and immunomodulatory properties [39,40,41]. Therefore, the ameliorative effects of CSO on CIA rats observed in this study are likely attributable to the synergistic actions of these key components. Our research demonstrates that a dosage of 8.4 g/kg of CSO significantly mitigates the symptoms of arthritis in a CIA rat model. Notably, treatment with CSO led to a reduction in the AI, paw swelling, and the expression of pro-inflammatory factors and bone destruction regulatory factors. Moreover, these doses resulted in improved pathological outcomes and imaging assessments of tissue damage, evidenced by reductions in CT bone parameters, HE staining, SOFG staining, and the number of osteoclasts present in the joint cavity. Collectively, these findings suggest that CSO not only alleviates clinical symptoms and pathological damage associated with arthritis but also holds promise as a therapeutic agent for RA. Important inflammatory mediators such as TNF-α, IL-6 and IL-1β play a key role in the inflammatory process of RA. TNF-α stimulates a variety of immune cells, causing them to release more cytokines, exacerbating the chain reaction of inflammation [42]. In contrast, IL-6 stimulates B cells to become active and promotes antibody production, leading to undesirable consequences such as thickening of the synovial membrane, cartilage damage and bone loss [43]. Similarly, IL-1β promotes the synthesis of inflammatory mediators in synovial cells, facilitating cartilage destruction and bone resorption, which further exacerbates the progression of RA [44]. In this study, we found that IL-1β, IL-6 and TNF-α levels in the serum of CIA rats were much higher than those of the CON group. In particular, after giving CSO, the levels of these inflammatory factors dropped significantly. In addition, CSO also reduced the expression of TNF-α and IL-6 in the ankle joints of CIA rats, where the inhibitory effect on the expression of inflammatory factors was strongest. These results suggest that CSO may have a good anti-inflammatory effect and may be able to slow the development of RA by affecting key inflammatory mediators.

An imbalance in intestinal flora is increasingly recognized as a factor contributing to the development of some chronic inflammatory joint diseases, including rheumatoid arthritis. In patients with RA, the intestinal microbiota often suffers from disorders, manifested by a decrease in the number of beneficial probiotics and an increase in harmful bacteria. This type of dysbiology is believed to exacerbate RA by stimulating the body’s innate immune defenses and activating the gut-joint axis, amplifying systemic inflammation and joint damage [45]. Recent studies have highlighted promising therapeutic strategies that aim to modulate the gut microbiota, including the use of traditional Chinese medicine. These interventions have shown potential as adjunctive treatments for RA by directly or indirectly restoring a healthier microbiota composition. For example, it was demonstrated that Sinomenine alleviated arthritis symptoms in CIA rats by promoting the enrichment of two beneficial *Lactobacillus* species, *L. paracasei* and *L. casei* [46]. Similarly, it was shown that licorice supplementation could regulate the composition of the gut microbiota, effectively controlling the “gut–joint axis” and mitigating the inflammatory responses in CIA rats [47]. In line with these findings, our study investigated the effects of CSO on gut microbiota in CIA rats. We observed that CSO treatment significantly enhanced both species richness and community diversity within the gut microbiota. Importantly, CSO administration led to an increase in the proportions of beneficial probiotics, including *Lactobacillus*, *Limosilactobacillus*, and *Romboutsia*. In contrast, CSO administration led to a decrease in the proportions of harmful bacterial taxa, including *Akkermansia*, *Bacteroides, Incertae Sedis*, and *UCG_005*. Furthermore, CSO treatment elevated the F/B ratio at the phylum level, indicative of a healthier microbiota balance. Notably, the beneficial strain *Lactobacillus* and *Limosilactobacillus* exhibited a negative correlation with the severity of RA symptoms with RA like AI, paw swelling, inflammatory factors (TNF-α, IL-1β, and IL-6), pathological scores (HE, SOFG, and osteoclast count). In contrast, the pathogenic strain *Bacteroides* and *Akkermansia* showed a positive correlation with the severity of arthritis symptoms. These findings suggest that modulation by CSO-improved strains may play a pivotal role in regulating inflammatory responses and bone damage, thereby highlighting potential therapeutic applications in managing immune-related disorders.

Short-chain fatty acids (SCFAs) are the main intermediates for beneficial effects of intestinal flora, and they are important in shaping the host’s immune function and inflammatory activity. If the intestinal flora is destroyed (i.e., dysbiosis), it will seriously affect the metabolism of SCFAs, especially the acetic, propionic, butyric and valeric acids. This disruption is recognized as a central factor in the initiation and progression of various chronic inflammatory diseases [48]. In healthy individuals, primary degraders, such as members of the phyla *Bacteroidetes* and *Firmicutes*, are responsible for breaking down complex polysaccharides into oligosaccharides and monosaccharides, which can subsequently be utilized by secondary metabolic bacteria. Through symbiotic mechanisms, including acetic acid-propionic acid-butyric acid cross-feeding, SCFAs are stably produced within the body. However, dysbiosis interferes with this delicate metabolic network, leading to a reduction in SCFAs-producing bacteria and an increase in protein-fermenting bacteria and potential pathogens. This imbalance results in a significant decrease in overall SCFAs levels, further compounding the pathogenesis of chronic inflammatory conditions [49]. SCFAs, particularly butyrate, have been shown to interact with the immune system via multiple mechanisms [50]. These metabolites modulate the production of pro-inflammatory cytokines such as TNF-α, IL-1β, IL-6, and nitric oxide, thereby contributing to the resolution of inflammation [51,52]. Consequently, a reduction in gut-derived butyrate not only impairs intestinal barrier function, allowing microbial products to translocate into the bloodstream, but also disrupts immune homeostasis, exacerbating both local and systemic inflammation [53]. In this study, non-targeted metabolomic analysis revealed disturbances in fatty acid metabolism in CIA rats, with targeted profiling identifying butyrate as a key metabolite. These disturbances are consistent with previous findings that SCFAs, particularly butyrate, play an important role in maintaining immune homeostasis and gut microbiota balance. In CIA model, serum butyrate levels were markedly depleted, indicating a possible link between disrupted fatty acid metabolism and the pathogenesis of RA. However, these levels were significantly restored following treatment with CSO, suggesting that CSO may alleviate RA symptoms by influencing butyrate metabolism. Notably, exogenous supplementation with butyrate was able to recapitulate the therapeutic effects of CSO, further validating butyrate’s pivotal role in the anti-RA mechanism of CSO. This finding aligns with the established role of butyrate in modulating immune responses, as it is known to exert anti-inflammatory effects through its influence on regulatory T cells and pro-inflammatory cytokine production. Furthermore, correlation analyses demonstrated that butyrate levels were positively associated with the abundance of beneficial probiotics, such as *Limosilactobacillus*, and inversely correlated with the severity of RA symptoms. These results indicate that butyrate may act as a key metabolite linking gut microbiota composition with systemic immune responses. Conversely, butyrate levels were negatively correlated with the abundance of potentially pathogenic bacteria, including *Bacteroides*, which is positively correlated with RA symptom severity. This suggests that the beneficial effects of butyrate may be mediated, in part, through its capacity to modulate the gut microbiome, promoting a healthier microbiota that supports immune regulation. Taken together, these findings suggest that butyrate serves not only as a metabolic mediator but also as a key player in the “gut–joint axis”, contributing to the therapeutic efficacy of CSO in RA.

In this study, we employed network pharmacology to identify NLRP3 as a key protein regulated by CSO in rheumatoid arthritis. We used network pharmacology and molecular docking methods to find that NLRP3 is a key target for CSO regulation of rheumatoid arthritis. The NLRP3 inflammasome is a multi-protein complex composed of pro-caspase-1, the NLRP3, and the ASC [27,54]. Numerous studies have underscored the critical involvement of the NLRP3 inflammasome in the pathogenesis of RA [29]. NLRP3 is a widely studied basic inflammatory body that is at the core of controlling inflammatory responses, the imbalance between catabolic and anabolic pathways, and the pathological remodeling of synovial tissue and chondrocytes in arthritis. Within the joint, NLRP3 interacts with ASC in the cytoplasm, leading to the recruitment of pro-caspase-1 and the subsequent formation of the NLRP3 inflammasome. This complex activates caspase-1, causing the cleavage of IL-1β and GSDMD. The result is that mature IL-1β and IL-18 are released, making inflammation more severe and RA disease worse. Furthermore, treatment with the NLRP3 inhibitor MCC950 for two weeks significantly reduced inflammation and bone destruction in CIA mice, further emphasizing the critical role of the NLRP3 inflammasome in RA pathogenesis [55]. In the present study, CIA resulted in a significant upregulation of the gene expression of NLRP3, ASC, Caspase-1, GSDMD, IL-1β, and IL-18 in the ankle joints of rats. Furthermore, the protein expression levels of NLRP3, Caspase-1, and IL-18 were also notably increased in the CIA group. These findings underscore the activation of the NLRP3 inflammasome in the progression of arthritis and provide further evidence of its role in driving inflammatory processes in RA pathology. Notably, treatment with CSO led to a substantial reduction in the expression of these genes and proteins. These findings suggest that CSO targets and inhibits NLRP3 inflammasome activation, thereby mitigating inflammation. This highlights the potential of CSO as a therapeutic agent for modulating inflammatory processes, offering promise in the treatment of RA.

Based on our findings that CSO supplementation reshaped the gut microbiota and elevated fecal butyrate levels, we speculate that the observed suppression of NLRP3 inflammasome activation and the subsequent attenuation of systemic inflammation may be mediated through the gut microbiota-butyrate axis. SCFAs derived from the intestinal flora might inhibit histone deacetylases and modify histone tails to regulate epigenetic modification and regulate G-protein-coupled receptor (GPCR) 41, GPCR 43, GPCR 109a and olfactory receptor 78 (Olfr 78) to play a biological role in inhibiting NLRP3 in mice [56]. The study showed that SCFAs might inhibit the NLRP3/ASC/Caspase-1 signaling pathway to improve inflammation [57]. Our subsequent research will be further refined that CSO enriches butyrate-producing bacteria, leading to increased butyrate levels to inhibit NLRP3 inflammasome activation via G-protein coupled receptor signaling, histone deacetylase inhibition, or enhancement of intestinal barrier integrity.

This study elucidates the mechanisms by which CSO alleviates RA, including gut microbiota remodeling, increased butyrate production, and inhibition of NLRP3 inflammasome activation, thereby reducing inflammation. Our findings demonstrate that CSO improves gut microbiota dysbiosis and significantly elevates the reduced butyrate levels in RA models. Network pharmacology and molecular docking analyses further support the involvement of NLRP3 inflammasome in CSO’s regulation of RA progression. The therapeutic effect of CSO comes from regulating the intestinal microbiota-butyric acid axis and reducing NLRP3 activation. This shows that CSO is expected to become a traditional herbal medicine for the treatment of RA. However, the precise mechanisms underlying CSO’s inhibition of NLRP3 inflammasome activation and its effects on gut microbiota require further investigation. Additionally, the efficacy of oral CSO supplementation in human RA patients warrants further clinical exploration.

This study has several limitations. The first concerns the mechanistic evidence. While our results provide compelling yet associative data that CSO ameliorates CIA by inhibiting NLRP3 inflammsome and modulating the gut microbiota-butyrate-joint axis, definitive proof of causality awaits interventional validation via methods such as fecal microbiota transplantation. The second limitation lies in the inherent complexity of CSO as a multi-compound formulation. Although we have identified potential active pathways, disentangling the precise contribution of individual components and their synergies remains a major challenge in the field of natural product research. Finally, the translational gap between pre-clinical models and human disease must be considered. Our findings in animal systems warrant further investigation in large-scale, randomized controlled clinical trials to firmly establish the efficacy and safety profile of CSO for treating RA in humans.

## 4. Materials and Methods

### 4.1. The Source of Materials

Coix Seed Oil (CSO) was purchased from Kanglaite Pharmaceutical Co., Ltd. (Hangzhou, China). The oil was extracted using Supercritical carbon dioxide fluid extraction (SFE-CO2), a solvent-free method that preserves the bioactive components of the oil. Methotrexate (MTX) was acquired from Xinyi Pharmaceutical Co., Ltd. (Shanghai, China). Chicken type II collagen, along with complete Freund’s adjuvant (CFA), was obtained from Chondrex (Woodinville, WA, USA). Commercial ELISA kits for the measurement of TNF-α, IL-6 and IL-1β levels were supplied by Guduo Biotechnology (Shanghai, China). ELISA kits for Butyric acid were from Hengyuan Biotechnology (Shanghai, China). Staining reagents, including Hematoxylin Eosin, Safranin O fast green and Tartar Phosphatase stains, were from Solarbio (Beijing, China). The Immunochromogenic reagent kit was obtained from MXB Biotechnologies (Fuzhou, China). Antibodies for NLRP3, Caspase-1, and IL-18 were purchased from Abcam (Cambridge, UK) and Affinity (Melbourne, Australia).

### 4.2. Preparation and UHPLC-MS Analysis for CSO Extract

Prior to UHPLC-OE/MS analysis, protein precipitation was performed to extract metabolites. Briefly, after thawing, samples were centrifuged at 4 °C and 12,000 rpm (RCF 13,800× *g*) for 15 min. A 300 μL aliquot of the supernatant was then mixed with 1000 μL of ice-cold extraction solvent (methanol:acetonitrile:water, 2:2:1, *v*/*v*/*v*) containing a mixture of isotope-labeled internal standards. The solution was vortexed vigorously for 30 s and subsequently sonicated in an ice-water bath for 5 min. To ensure complete protein precipitation, the mixture was incubated at −20 °C for 1 h, followed by centrifugation again under the same conditions (4 °C, 12,000 rpm for 15 min). The final supernatant was filtered through a 0.22 μm nylon membrane prior to UHPLC-MS/MS analysis.

Chromatographic separation was achieved on a Vanquish UHPLC system (Thermo Fisher Scientific, Waltham, MA, USA) using a Phenomenex Kinetex C18 column (2.1 mm × 100 mm, 2.6 μm) maintained at 4 °C. The mobile phase consisted of (A) water with 0.01% acetic acid and (B) a mixture of isopropanol and acetonitrile (1:1, *v*/*v*). The injection volume was 2 μL. Mass spectrometric detection was carried out on an Orbitrap Exploris 120 instrument (Thermo Fisher Scientific) controlled by Xcalibur software (version 4.4). The instrument was operated in data-dependent acquisition (DDA) mode. Key MS parameters were set as follows: sheath gas flow rate, 50 Arb; auxiliary gas flow rate, 15 Arb; capillary temperature, 320 °C; full MS resolution, 60,000; MS/MS resolution, 15,000; stepped normalized collision energy (SNCE), 20/30/40; spray voltage, +3.8 kV (positive ion mode) or −3.4 kV (negative ion mode).

### 4.3. Animals Experimental Design

A total of 36 male SD rats (6 weeks old, 180–220 g, N = 36) and 30 DBA/1 mice (5 weeks old, N = 30) were housed under specific pathogen-free conditions, sourced from HFK Biotechnology Co. (License No. SCXK [Beijing] 2019-0008; animal qualification certificate No. 110322231103515862). The animals are kept in a particularly clean, specific pathogen free (SPF) area where the environment is strictly controlled: temperatures are kept between 22 and 26 °C, humidity between 40% and 60% with a 12 h light/12 h dark cycle. All experimental procedures fully comply with the National Institutes of Health (NIH) Guidelines for the Care and Use of Laboratory Animals. The study was approved by the China Medical University Ethics Committee (Ethical Approval No. CMU20231436, 4 December 2023). The sample size was determined by a power analysis to ensure sufficient power to detect a significant effect. The doses were adjusted based on the application of the standard equivalent dose conversion coefficient method across species.

Rats were randomly divided into six groups (*n* = 6): normal control (CON, 0.9% saline), model (CIA, 0.9% saline), CSO low-dose (CSO-L, 2.1 g/kg/d), CSO medium-dose (CSO-M, 4.2 g/kg/d), CSO high-dose (CSO-H, 8.4 g/kg/d), and methotrexate (MTX, 0.5 mg/kg, three times a week). The experimental timeline is shown in Figure 1A.

Mice were randomly divided into five groups (*n* = 6): normal control (CON, 0.9% saline), model (CIA, 0.9% saline), CSO low-dose (CSO-L, 3.12 g/kg/d), CSO high-dose (CSO-H, 12.48 g/kg/d), and butyric acid (BA, 100 mg/kg/d). The experimental timeline is shown in Figure 7A.

The collagen-induced arthritis model was generated following a well-established protocol [58]. Rats and mice, except for the control group, received a subcutaneous administration of the collagen adjuvant at a dosage of 0.1 mL per 100 g of body weight (chicken type II collagen and CFA emulsified) at multiple sites on the tail base (2 cm from the tail) and back (3–5 points). The control group received 0.9% saline at the same sites. The animals received the initial and booster immunizations on days 0 and 7, respectively. Starting on Day 8, rats and mice received saline or the designated treatments orally for 28 consecutive days. On Day 35, rats and mice were euthanized for further analysis.

### 4.4. Evaluation of Arthritis in Rats and Mice

A vernier caliper was used to measure the width of the rear of the left ankle joint to determine the extent of arthritis. Weekly evaluations of body weight and paw thickness were conducted to systematically monitor and record any changes over time. The Arthritis Index (AI) was assigned on a five-point scale for each limb, thereby establishing an upper limit of 16 points for individual rodents [59]. A 0–4 point scale was implemented, with scores defined as: 0, asymptomatic; 1, localized toe joint swelling; 2, mild symptoms from toe to ankle; 3, moderate manifestations below the ankle; and 4, severe presentation including redness, swelling, joint dysfunction, deformity, and potential ulceration or bleeding.

### 4.5. Enzyme-Linked Immunosorbent Assay (ELISA)

Following centrifugation at 2000 rpm for 15 min to isolate serum, the concentrations of butyric acid and several pro-inflammatory mediators (TNF-α, IL-6, IL-1β) were subsequently quantified using commercial ELISA kits.

### 4.6. Micro-CT

Micro-CT scanning of fixed ankle joints was performed under specified conditions for 3D structural analysis, with subsequent architectural parameters (BV, BV/TV, BS/TV) being quantified in triplicate using CTAn software (https://www.ctan.org/, accessed on 18 December 2025).

### 4.7. Immunohistochemical (IHC) Staining

Immunohistochemical staining for TNF-α and IL-6 (1:400) was performed on deparaffinized cartilage sections with DAB visualization and hematoxylin counterstaining, followed by light microscopic examination and semi-quantitative analysis using ImageJ software (https://imagej.net/downloads, accessed on 18 December 2025).

### 4.8. Hematoxylin-Eosin (HE) Staining

After fixation, decalcification, paraffin-embedding, and HE staining of rodent ankle joints, histological alterations including cartilage morphology, inflammatory cell infiltration, vascularization, and synovial hyperplasia were assessed.

### 4.9. Safranin O Fast Green (SOFG) Staining

Following dewaxing, rehydration, and a water rinse, tissue sections were sequentially stained with 2% Fast Green and 0.1% Ponceau S, then dehydrated and mounted for microscopic examination, which differentiated cartilage and connective tissues (red) from bone (green).

### 4.10. Tartrate-Resistant Acid Phosphatase (TRAP) Staining

Dewaxed and dehydrated paraffin sections were processed for TRAP staining at 37 °C and hematoxylin counterstaining, with osteoclasts identified as cells with wine-red cytoplasm and light green/blue nuclei.

### 4.11. Immunofluorescence (If) Staining

Dewaxed and rehydrated joint sections were processed for antigen retrieval and blocking, incubated sequentially with specified primary antibodies and a DyLight 488-conjugated secondary antibody, and finally counterstained with DAPI for fluorescence microscopy observation.

### 4.12. Fecal Metabolic Analysis

Frozen intestinal content (~60 mg) was homogenized with steel beads in 600 μL of cold methanol-water (4:1, *v*/*v*) containing 20 μL of internal standard (L-2-chlorophenylalanine, 0.3 mg/mL) after 5 min at −20 °C. The mixture was ultrasonicated (ice-water bath, 10 min), incubated at −20 °C for 30 min, and centrifuged (13,000 rpm, 4 °C, 10 min). A 300 μL aliquot of supernatant was dried, reconstituted in 400 μL methanol-water (1:4, *v*/*v*), vortexed, sonicated, and incubated at −20 °C for 2 h. After a final centrifugation, 150 μL of supernatant was filtered (0.22 μm) into an LC vial and stored at −80 °C until analysis. A pooled QC sample was prepared from all extracts. All solvents were pre-chilled to −20 °C. LC-MS analysis was performed using an AB ExionLC UPLC system coupled to a Q Exactive mass spectrometer (Thermo Fisher Scientific, Waltham, MA, USA). Separation used an ACQUITY UPLC HSS T3 column (100 × 2.1 mm, 1.8 μm) at 45 °C, with mobile phase A (water with 0.1% formic acid) and B (acetonitrile with 0.1% formic acid) at 0.35 mL/min. Injection volume was 2 μL. ESI source parameters were: spray voltages (positive/negative ion modes), sheath and auxiliary gas settings, and capillary temperature as per instrument method. Data were acquired in full scan mode.

### 4.13. SCFAs Quantification in Serum

The SCFAs in plasma samples, including acetic acid, butyric acid, isovaleric acid and pentanoic acid were comprehensively analyzed. Target metabolites were qualitatively and quantitatively analyzed using a UPLC-ESI-MS/MS system. Chromatographic separation was performed on an Acquity UPLC BEH C18 column (2.1 × 100 mm, 1.7 µm) maintained at 40 °C. The mobile phase consisted of 0.1% formic acid in water (solvent A) and acetonitrile (solvent B), delivered at a flow rate of 0.35 mL/min with an injection volume of 5 µL. The gradient elution program was as follows: 0–1 min, 90% A; 1–2 min, 90% to 75% A; 2–6 min, 75% to 65% A; 6–6.5 min, 65% to 5% A; 6.5–7.8 min, hold at 5% A; 7.8–7.81 min, 5% to 90% A; 7.81–8.5 min, hold at 90% A. Mass spectrometric detection was conducted using an electrospray ionization (ESI) source. The ion source temperature was set at 450 °C, with spray voltages of +5500 V and −4500 V for positive and negative ion modes, respectively. The curtain gas pressure was 35 psi, the nebulizing gas (Gas1) pressure was 50 psi, and the auxiliary heating gas (Gas2) pressure was 60 psi. The collision-activated dissociation (CAD) parameter was set to “Medium”.

### 4.14. 16S rRNA Sequencing Analysis

Alpha diversity was assessed using Simpson, Faith’s PD, and Shannon indices to evaluate genus richness and evenness across samples. Differentially abundant microbial features were identified through Linear Discriminant Analysis Effect Size (LEfSe), applying a threshold of LDA score >3 and *p* < 0.05. Cluster analysis was performed to group samples based on microbial composition and functional similarities, revealing distribution patterns. Principal Coordinate Analysis (PCoA) utilizing UniFrac distance matrices was conducted to assess differences in microbial community structure and visualize sample distribution in a low-dimensional space.

### 4.15. Network Pharmacology Analysis

Potential targets of Coix lacryma-jobi active ingredients were identified by screening databases such as TCMSP. Canonical SMILES structures and molecular weights were obtained from PubChem. These structures were then used to predict potential targets via Swiss Target Prediction and Target Net. Potential targets for rheumatoid arthritis (RA) were concurrently obtained from the Gene Cards and OMIM databases. Subsequently, the common targets shared with the medicine components were screened by constructing a Venn diagram. For these overlapping targets, a protein–protein interaction (PPI) network was generated using the STRING database (v12.0) and visualized with Cytoscape 3.9.1. Finally, their biological functions and pathways were investigated through enrichment analyses based on the Gene Ontology (GO) and the Kyoto Encyclopedia of Genes and Genomes (KEGG).

### 4.16. Molecular Docking Analysis

The main reagents (Coixenolide, Sitosterol, and Stigmasterol) and the NLRP3 protein (PDB ID: 7ALV) were analyzed using AutoDock Vina v1.2.3, following ligand and receptor preparation, with subsequent analysis and visualization of interactions conducted in Discovery Studio (https://discover.3ds.com/discovery-studio-visualizer-download, accessed on 18 December 2025) and PyMOL (https://pymol.org/, accessed on 18 December 2025).

### 4.17. Real-Time (RT)-qPCR

Joint tissues were rapidly collected and placed on ice or in RNAlater. After rinsing with saline, the tissue was minced into small pieces. For soft tissues, homogenization was performed in Trizol using a tissue homogenizer (60 Hz, 45–60 s, repeated 3 times). For hard or calcified tissues, samples in Trizol were first incubated at 4 °C for 3 h, followed by grinding in an ice-water bath or controlled ultrasonic disruption. RNA was then extracted by adding chloroform, centrifuging, precipitating with isopropanol at −20 °C, washing with 75% ethanol, and finally dissolving in RNase-free water. All steps were performed on ice or at 4 °C using RNase-free reagents and consumables to prevent degradation. The recommended tissue amount was 50–100 mg per sample. RNA quality and concentration were verified by spectrophotometry and gel electrophoresis prior to qPCR. The cDNA was subsequently diluted and incorporated into the prepared amplification system, with amplification conducted according to the kit guidelines. Based on the 2^−ΔΔCt^ method, the relative quantification of target gene mRNA expression was determined, with the corresponding primer sequences detailed in Table 3.

### 4.18. Statistical Analysis

The data are reported as means ± standard deviation. As extreme values in the dataset compromised statistical accuracy, data processing was conducted to ensure a minimum of three samples per group for analysis. GraphPad Prism 10.0 (GraphPad Software Inc., San Diego, USA) was utilized for statistical computations and data visualization, with supplementary figure editing performed on Wekemo Bioincloud. The significance of differences among groups was determined by one-way ANOVA, supplemented with Fisher’s LSD post hoc analysis. Statistical significance was set at *p* < 0.05.

## 5. Conclusions

In conclusion, our study demonstrates that CSO can effectively reduce joint inflammation and bone damage in CIA rats. This therapeutic effect is mediated through the modulation of the gut microbiota composition and abundance, particularly by increasing the population of bacteria that produce short-chain fatty acids, such as butyrate. As a result, CSO treatment leads to elevated blood butyrate levels. We propose that CSO alleviates inflammation by regulating the level of butyrate that short-chain fatty acid metabolite produced from the gut microbiota and by inhibiting NLRP3 inflammasome activation. This dual mechanism ultimately reduces both joint inflammation and bone destruction. Our findings offer new insights into the therapeutic action of CSO in RA, specifically highlighting its modulation of the “gut-butyrate-joint” axis.

## Figures and Tables

**Figure 1 pharmaceuticals-19-00048-f001:**
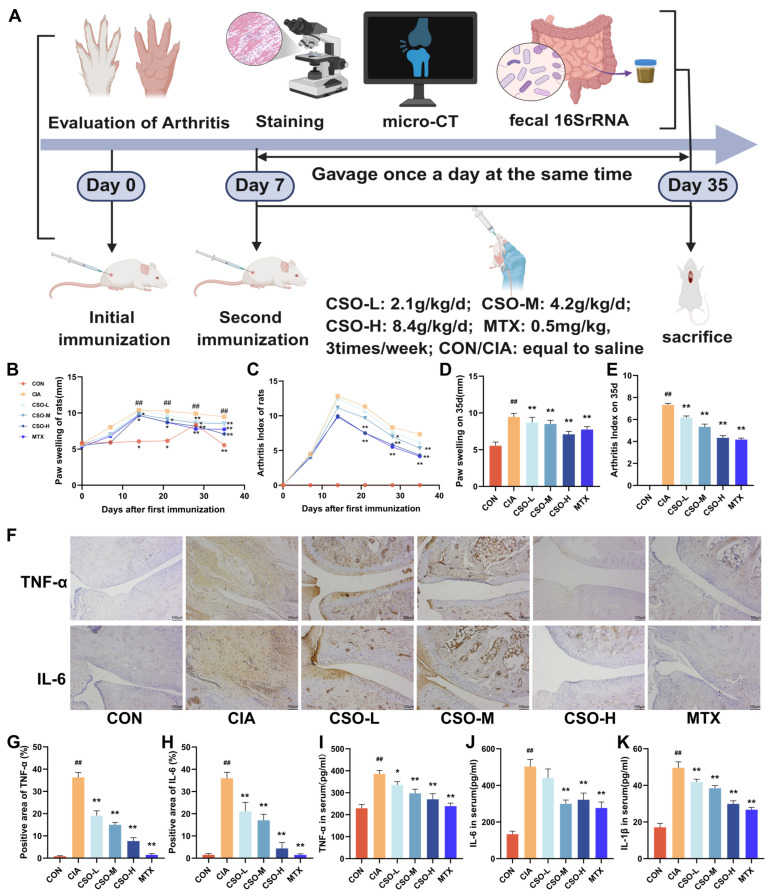
CSO relieved arthritic symptoms and reduced inflammatory markers in CIA rats. (**A**) Experimental design flowchart. (**B**) Paw swelling changes. (**C**) AI scores changes. (**D**) Paw swelling on day 35. (**E**) AI on day 35. (**F**) Immunohistochemical staining of TNF-α and IL-6 in ankle joints (*n* = 3). (**G**,**H**) Quantitative analysis of TNF-α and IL-6 in ankle joints (*n* = 3). (**I**–**K**) Serum levels of TNF-α, IL-6, and IL-1β (*n* = 6). Data are expressed as mean ± SD. ^##^ *p* < 0.01, compared with CON; * *p* < 0.05, ** *p* < 0.01, compared with CIA.

**Figure 2 pharmaceuticals-19-00048-f002:**
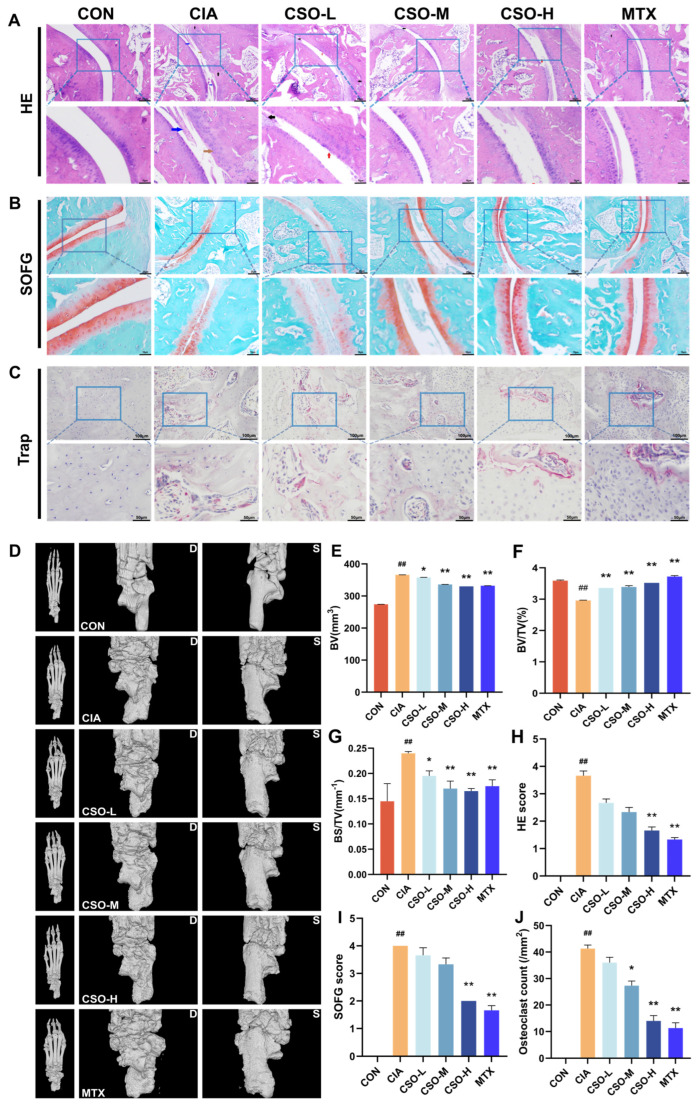
Effect of CSO on joint pathology and imaging in CIA rats. (**A**) HE staining of the ankle joint (red arrow: synovial hyperplasia, black arrow: inflammatory cell infiltration, blue arrow: pannus formation). (**B**) SOFG staining of the ankle joint. (**C**) TRAP staining of the ankle joint. (**D**) CT scan of the paw (Dorsum of the foot-D: left; Sole of the foot-S: right). (**E**) BV (mm^3^). (**F**) BV/TV (%). (**G**) BS/TV (mm^−1^). (**H**) HE staining score. (**I**) SOFG score. (**J**) Quantification of bone-positive cells in the ankle joints. Data are presented in mean ± SD. ^##^ *p* < 0.01, compared with CON; * *p* < 0.05, ** *p* < 0.01, compared with CIA; *n* = 3.

**Figure 3 pharmaceuticals-19-00048-f003:**
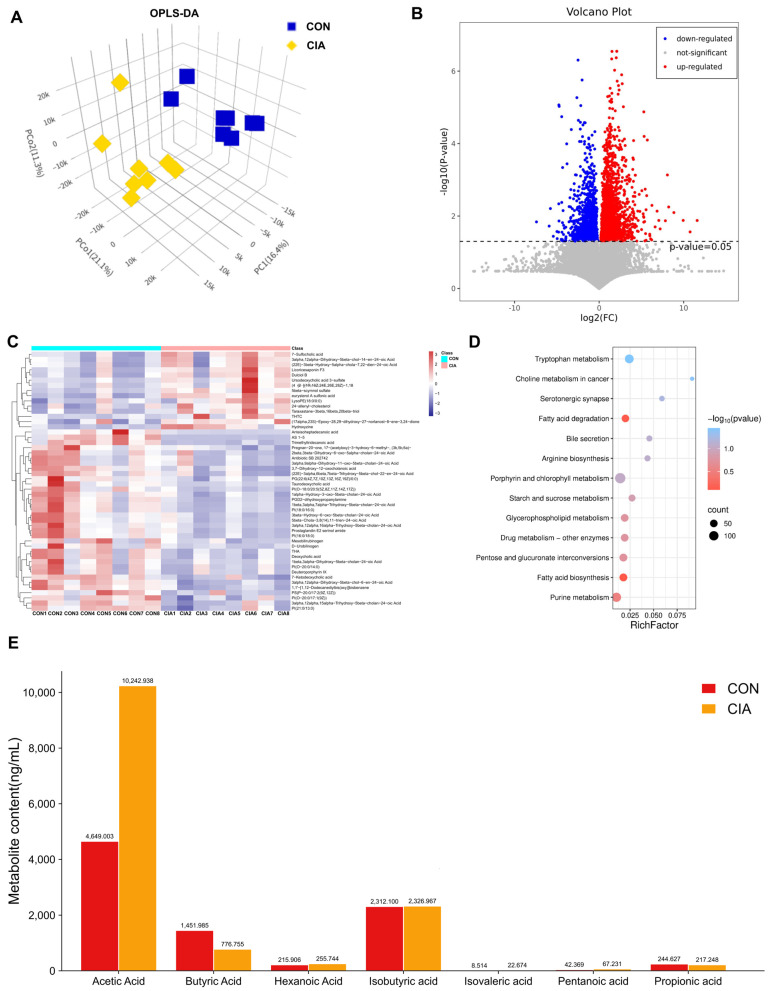
Significantly differentially expressed metabolites. (**A**) OPLS-DA of fecal metabolites (*n* = 8). (**B**) Volcano plot of differential metabolites (CON vs. CIA, *n* = 8). (**C**) Heatmap of top 50 differential metabolites (*n* = 8). (**D**) KEGG pathway enrichment analysis (*n* = 8). (**E**) Serum SCFAs concentrations (*n* = 6). Data are expressed as mean ± SD.

**Figure 4 pharmaceuticals-19-00048-f004:**
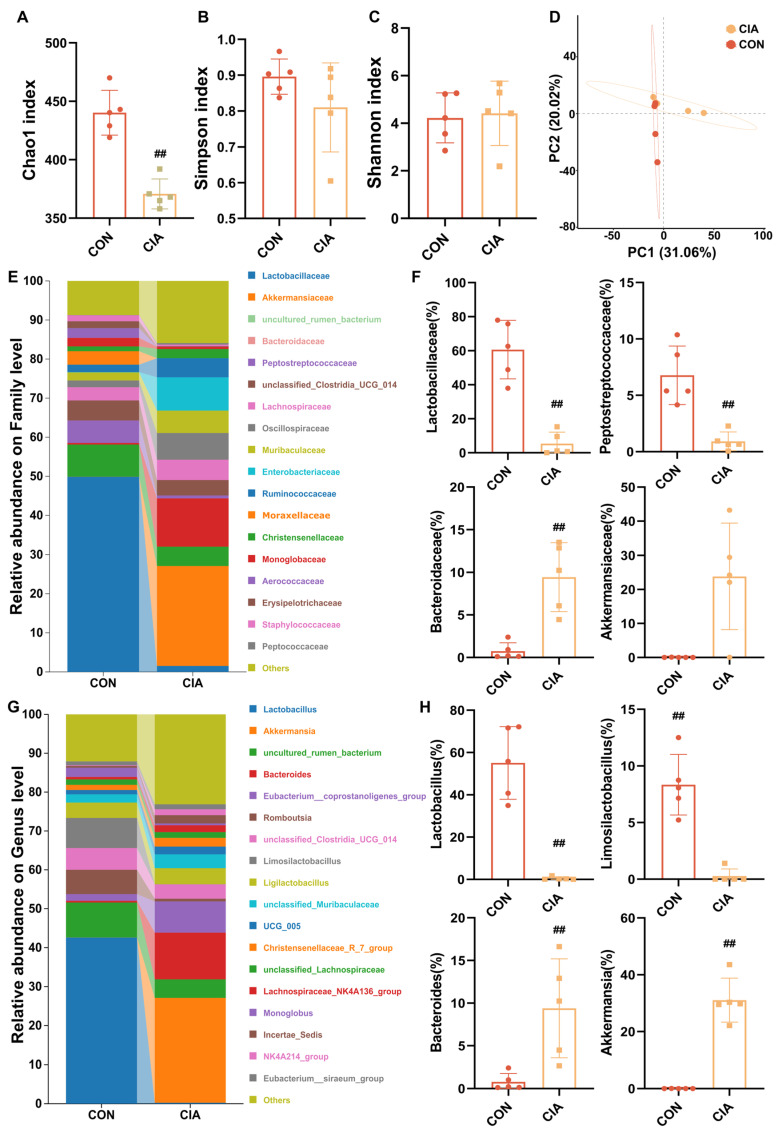
Dysregulated metabolism of gut microbiota in CIA rats. (**A**) Chao1 index (*n* = 5). (**B**) Simpson index (*n* = 5). (**C**) Shannon index (*n* = 5). (**D**) β-diversity of the gut microbiota based on PCA analysis (*n* = 5). (**E**) Percent taxa at the family level (*n* = 6). (**F**) Relative abundance of taxa at the family level (*n* = 5). (**G**) Percent taxa at the genus level (*n* = 6). (**H**) Relative abundance of taxa at the genus level (*n* = 5). Data are presented in mean ± SD. ^##^ *p* < 0.01, compared with CON.

**Figure 5 pharmaceuticals-19-00048-f005:**
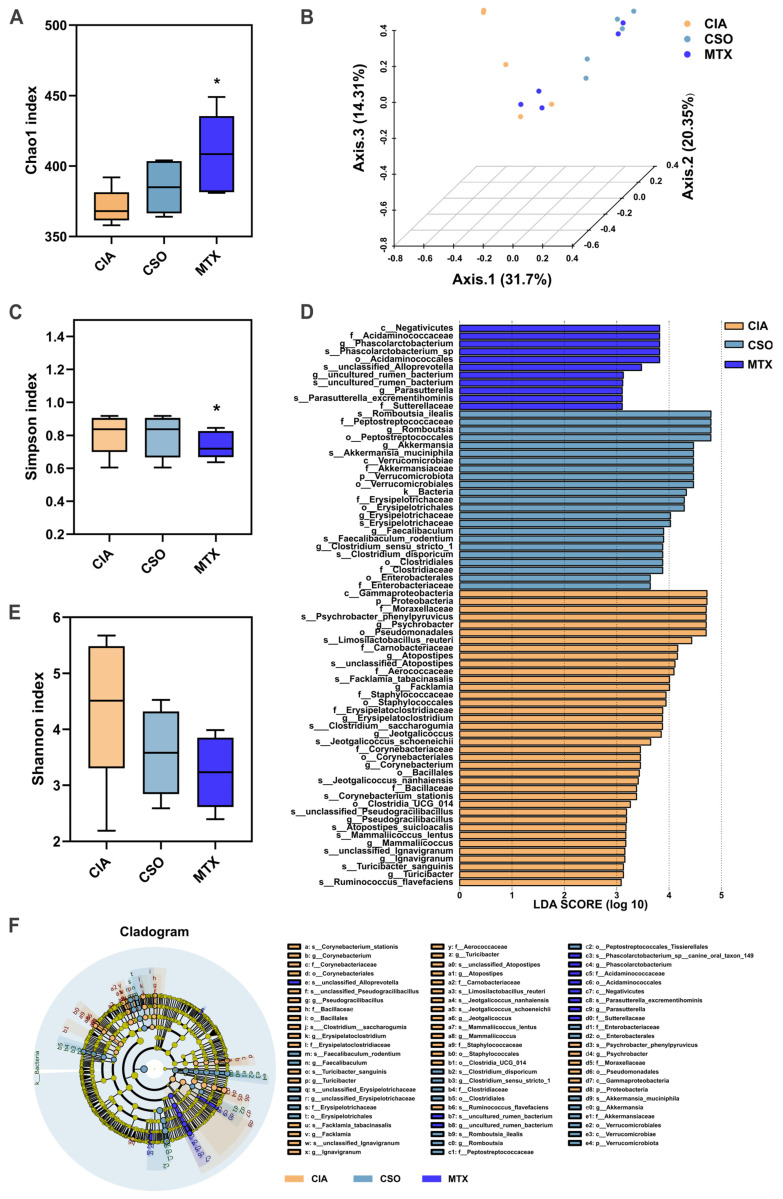
CSO regulated gut microbial diversity in CIA rats. (**A**) Chao1 index. (**B**) β-diversity of the gut microbiota based on PCoA analysis. (**C**) Simpson index. (**D**) LEfSe analysis of LDA score. (**E**) Shannon index. (**F**) LEfSe analysis cladogram. Data are presented in mean ± SD. * *p* < 0.05, compared with CIA; *n* = 5.

**Figure 6 pharmaceuticals-19-00048-f006:**
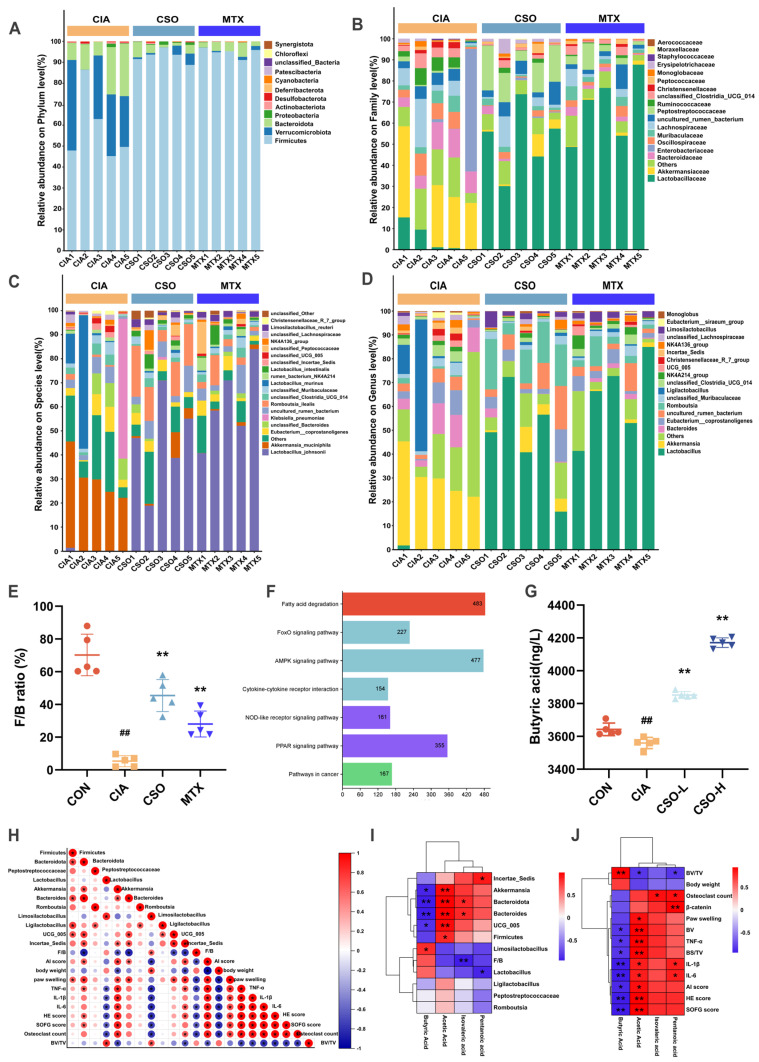
CSO modulation of gut microbiota-butyric acid. (**A**) Abundance analysis at the phylum level. (**B**) Abundance analysis at the family level. (**C**) Abundance analysis at the species level. (**D**) Abundance analysis at the genus level. (**E**) F/B ratio. (**F**) Predictive analysis of KEGG pathways. (**G**) Serum levels of butyric acid in mice. (**H**) Correlation analysis between gut microbiota, joint inflammation, and bone destruction. (**I**) Correlation analysis between SCFAs and gut microbiota. (**J**) Correlation analysis between SCFAs and inflammatory factors. Data are presented as mean ± SD. ^##^ *p* < 0.01, compared with CON; * *p* < 0.05, ** *p* < 0.01, compared with CIA; *n* = 5. When the correlation coefficient exceeds zero (*r* > 0), it reflects a positive association; conversely, a value less than zero (*r* < 0) indicates a negative association.

**Figure 7 pharmaceuticals-19-00048-f007:**
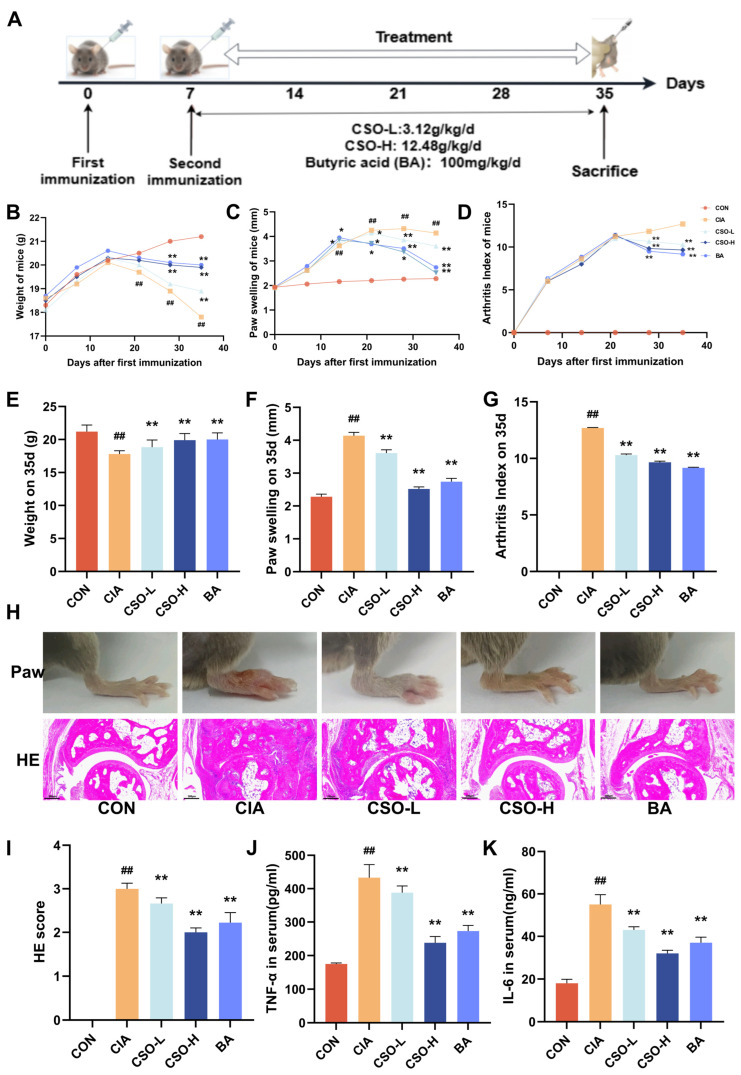
BA relieved arthritic symptoms in CIA mice. (**A**) Experimental design flowchart. (**B**) Body weight changes (*n* = 6). (**C**) Paw swelling changes (*n* = 6). (**D**) AI scores changes. (**E**) Body weight on day 35 (*n* = 6). (**F**) Paw swelling on day 35 (*n* = 6). (**G**) AI on day 35 (*n* = 6). (**H**) Images of the right hind paw of a mouse on the 35th day and HE staining of the ankle joint (*n* = 3). (**I**) HE staining score (*n* = 3). (**J**,**K**) Serum levels of TNF-α and IL-6 (*n* = 6). Data are expressed as mean ± SD. ^##^ *p* < 0.01, compared with CON; * *p* < 0.05, ** *p* < 0.01, compared with CIA.

**Figure 8 pharmaceuticals-19-00048-f008:**
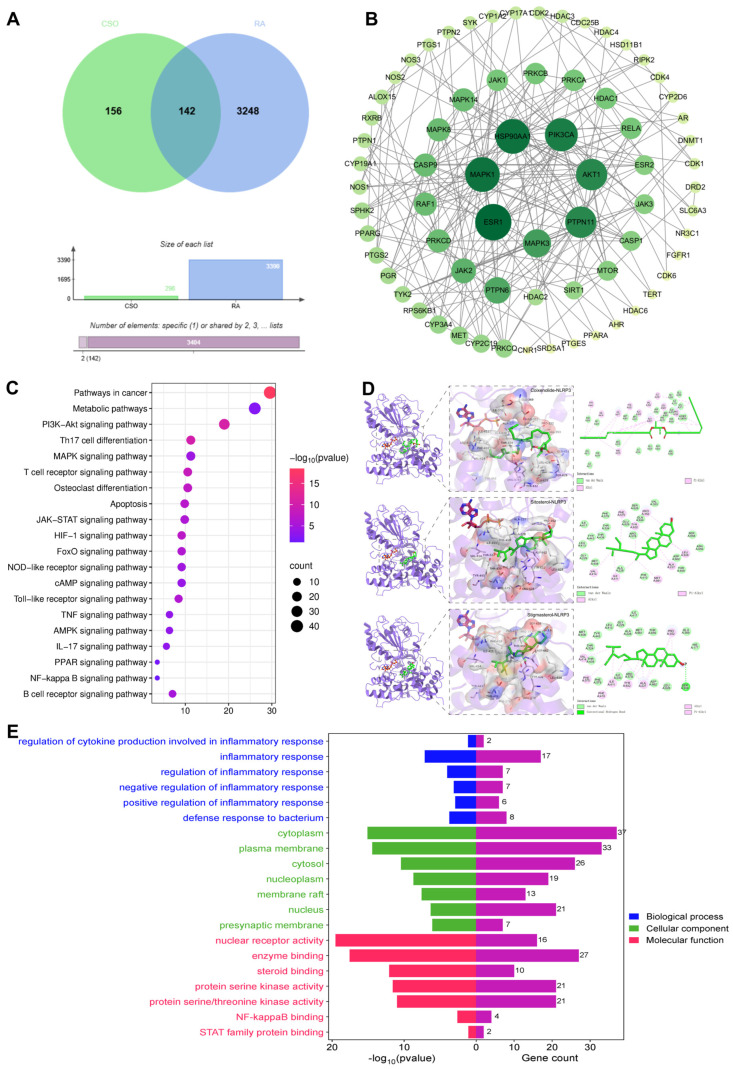
NLRP3 inflammasome may be a potential target for CSO in RA. (**A**) Venny diagram. (**B**) PPI network. (**C**) KEGG pathway enrichment analysis. (**D**) Molecular docking analysis of CSO with NLRP3. (**E**) GO function analysis of biological processes. Data are expressed as mean ± SD; *n* = 6.

**Figure 9 pharmaceuticals-19-00048-f009:**
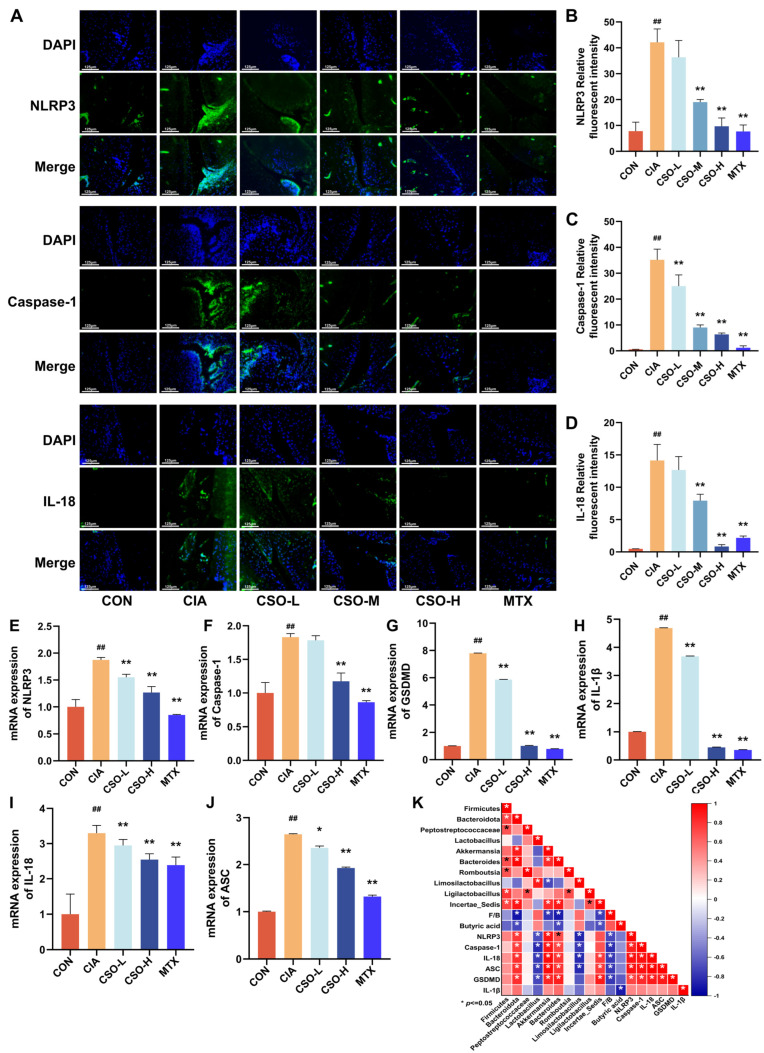
Effect of CSO on NLRP3 Inflammasome Activation. (**A**) Representative immunofluorescence images of NLRP3, Caspase-1 and IL-18. (**B**–**D**) Quantitative analysis of NLRP3, Caspase-1 and IL-18 in CIA rats. (**E**–**J**) The relative mRNA expressions of NLRP3, Caspase-1, GSDMD, IL-1β, IL-18 and ASC in CIA mice. (**K**) Correlation analysis between NLRP3 and gut microbiota. Data are presented as mean ± SD. ^##^ *p* < 0.01, compared with CON; * *p* < 0.05, ** *p* < 0.01, compared with CIA; *n* = 3. When the correlation coefficient exceeds zero (*r* > 0), it reflects a positive association; conversely, a value less than zero (*r* < 0) indicates a negative association.

**Table 1 pharmaceuticals-19-00048-t001:** Positive ingredients of CSO identified by UHPLC-OE/MS.

No.	Composite Scores	Compound	Formula	rt (s)	m/z
1	3.99	Indirubin	C_16_H_10_N_2_O_2_	429.4	263.0814
2	3.98	11-Dehydrocorticosterone	C_21_H_28_O_4_	354.5	345.2058
3	3.98	Cholic acid	C_24_H_40_O_5_	437.3	391.2842
4	3.98	Creatine	C_4_H_9_N_3_O_2_	46.7	132.0766
5	3.98	Galactose	C_6_H_12_O_6_	44.9	203.0525
6	3.98	Indigo	C_16_H_10_N_2_O_2_	408.8	263.0815
7	3.98	Piperine	C_17_H_19_NO_3_	416.5	286.1437
8	3.98	Proline	C_5_H_9_NO_2_	46.7	116.0705
9	3.98	Sph (t18:0)	C_18_H_39_NO_3_	408.6	318.3001
10	3.97	Tagatose	C_6_H_12_O_6_	44.9	203.0525

**Table 2 pharmaceuticals-19-00048-t002:** Negative ingredients of CSO identified by UHPLC-OE/MS.

No.	Composite Scores	Compound	Formula	rt (s)	m/z
1	3.99	Linoleic acid	C_18_H_32_O_2_	553.2	279.2331
2	3.98	Alanine	C_3_H_7_NO_2_	44.6	88.0405
3	3.98	alpha-Linolenic acid	C_18_H_30_O_2_	536.4	277.2175
4	3.98	cis-9-Palmitoleic acid	C_16_H_30_O_2_	548.2	253.2175
5	3.98	Myristic acid	C_14_H_28_O_2_	545.2	227.2019
6	3.98	Oleic acid	C_18_H_34_O_2_	570.8	281.2488
7	3.98	Sarcosine	C_3_H_7_NO_2_	44.6	88.0405
8	3.98	Sebacic acid	C_10_H_18_O_4_	357.7	201.1133
9	3.98	trans-Vaccenic acid	C_18_H_34_O_2_	570.8	281.2488
10	3.98	Undecanoic acid	C_11_H_22_O_2_	497.7	185.1549

**Table 3 pharmaceuticals-19-00048-t003:** Primer Sequences.

Name	Sequences (5′~3′)
NLRP3-F	ATGACTTTCCAGGAGTTCTTCGC
NLRP3-R	CCAAAGAGGAATCGGACAACAA
Caspase1-F	GGCTGACAAGATCCTGAGGG
Caspase1-R	TAGGTCCCGTGCCTTGTCC
ASC-F	CAGCACAGGCAAGCACTCATT
ASC-R	TCATCTTGTCTTGGCTGGTGG
GSDMD-F	GAGCTTTATGCTTGAAGGGTGA
GSDMD-R	ATGGAACAAAGCGCAGCAA
IL-1β-F	AGGCTCCGAGATGAACAACAAA
IL-1β-R	GTGCCGTCTTTCATTACACAGGA
IL-18-F	AATGGAGACCTGGAATCAGACAA
IL-18-R	GGTCACAGCCAGTCCTCTTACTTC
GAPDH-F	CCTCGTCCCGTAGACAAAATG
GAPDH-R	TGAGGTCAATGAAGGGGTCGT

## Data Availability

The original contributions presented in this study are included in the article/Appendix A. Further inquiries can be directed to the corresponding author.

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
