# Peer review of "Integrative Multi-Omics Elucidates the Therapeutic Effect of Coix Seed Oil on Rheumatoid Arthritis via the Gut-Butyrate-Joint Axis and NLRP3 Inflammasome Suppression"

_pharmaceuticals, 2025, doi:10.3390/ph19010048_

Round 1
Reviewer 1 Report
Comments and Suggestions for Authors
Attached

Reviewer 2 Report
Comments and Suggestions for Authors
The manuscript entitled, "Integrative Multi-Omics Elucidates the Therapeutic Effect of Coix Seed Oil on Rheumatoid Arthritis via the Gut-Butyrate-Joint Axis and NLRP3 Inflammasome Suppression" is an interesting study and highlights the importance of a traditional remedy of RA. The mechanistic approach for the therapeutic potential of Coix seed oil was evaluated in detail.
The result section of abstract lacks numerical results and this information is extremely important to quantify the therapeutic potential.
Introduction: Please elaborate the role of short chain fatty acids in RA, using support available from the recent literature.
Please also add what is in the oil that is responsible for therapeutic potential.
Please determine the chemical composition of oil and correlated the active components with the claimed therapeutic effect. Without chemical characterization, it is not appropriate to report the therapeutic potential.
In method LC-MS method is not appropriately described, please add the specifications of column, MS conditions and target range. Moreover, in results include the table showing composition and add in discussion the main components responsible for RA remedy.
The result and discussion sections are appropriate and well presented.
Overall the manuscript contains significant contribution, however above-mentioned points should be addressed.
Author Response
Please see the attachment.
|
Response to Reviewer 2 Comments
|
||||||||||||||||||||||||||||||||||||||||||||||||||||||||||||||||||||||||||||||||||||||||||||||||||||||||||||||||||||||||||||||||||||||
|
1. Summary |
|
|
||||||||||||||||||||||||||||||||||||||||||||||||||||||||||||||||||||||||||||||||||||||||||||||||||||||||||||||||||||||||||||||||||||
|
We sincerely appreciate the time and effort you dedicated to reviewing our manuscript and your insightful comments. In response, we have revised the manuscript with careful consideration to enhance its clarity, accessibility, and impact. Below, please find our point-by-point responses (highlighted in purple), with all corresponding changes marked in red within the revised manuscript. We believe these revisions have fully addressed the issues you raised.
|
||||||||||||||||||||||||||||||||||||||||||||||||||||||||||||||||||||||||||||||||||||||||||||||||||||||||||||||||||||||||||||||||||||||
|
2. Questions for General Evaluation |
Reviewer’s Evaluation |
Response and Revisions |
||||||||||||||||||||||||||||||||||||||||||||||||||||||||||||||||||||||||||||||||||||||||||||||||||||||||||||||||||||||||||||||||||||
|
Does the introduction provide sufficient background and include all relevant references? |
Can be improved |
Please see as described below. |
||||||||||||||||||||||||||||||||||||||||||||||||||||||||||||||||||||||||||||||||||||||||||||||||||||||||||||||||||||||||||||||||||||
|
Is the research design appropriate? |
Can be improved |
Please see as described below. |
||||||||||||||||||||||||||||||||||||||||||||||||||||||||||||||||||||||||||||||||||||||||||||||||||||||||||||||||||||||||||||||||||||
|
Are the methods adequately described? |
Can be improved |
Please see as described below. |
||||||||||||||||||||||||||||||||||||||||||||||||||||||||||||||||||||||||||||||||||||||||||||||||||||||||||||||||||||||||||||||||||||
|
Are the results clearly presented? |
Can be improved |
Please see as described below. |
||||||||||||||||||||||||||||||||||||||||||||||||||||||||||||||||||||||||||||||||||||||||||||||||||||||||||||||||||||||||||||||||||||
|
Are the conclusions supported by the results? |
Yes |
Please see as described below. |
||||||||||||||||||||||||||||||||||||||||||||||||||||||||||||||||||||||||||||||||||||||||||||||||||||||||||||||||||||||||||||||||||||
|
Are all figures and tables clear and well-presented? |
Yes |
Please see as described below. |
||||||||||||||||||||||||||||||||||||||||||||||||||||||||||||||||||||||||||||||||||||||||||||||||||||||||||||||||||||||||||||||||||||
|
3. Point-by-point response to Comments and Suggestions for Authors |
||||||||||||||||||||||||||||||||||||||||||||||||||||||||||||||||||||||||||||||||||||||||||||||||||||||||||||||||||||||||||||||||||||||
|
Comments 1: The manuscript entitled, "Integrative Multi-Omics Elucidates the Therapeutic Effect of Coix Seed Oil on Rheumatoid Arthritis via the Gut-Butyrate-Joint Axis and NLRP3 Inflammasome Suppression" is an interesting study and highlights the importance of a traditional remedy of RA. The mechanistic approach for the therapeutic potential of Coix seed oil was evaluated in detail. |
||||||||||||||||||||||||||||||||||||||||||||||||||||||||||||||||||||||||||||||||||||||||||||||||||||||||||||||||||||||||||||||||||||||
|
Response 1: We sincerely thank you for your positive and encouraging comments on our manuscript, and for acknowledging the interest and importance of our work in elucidating the mechanistic basis of Coix seed oil for rheumatoid arthritis treatment. We are very pleased that the reviewer found our integrative multi-omics and mechanistic approach to be detailed and compelling. We are grateful for the opportunity to share our findings with the scientific community and believe that delineating the role of CSO in modulating the gut-butyrate-joint axis and suppressing the NLRP3 inflammasome provides a valuable foundation for future research and development of novel therapeutic strategies for RA. |
||||||||||||||||||||||||||||||||||||||||||||||||||||||||||||||||||||||||||||||||||||||||||||||||||||||||||||||||||||||||||||||||||||||
|
Comments 2: The result section of abstract lacks numerical results and this information is extremely important to quantify the therapeutic potential. |
||||||||||||||||||||||||||||||||||||||||||||||||||||||||||||||||||||||||||||||||||||||||||||||||||||||||||||||||||||||||||||||||||||||
|
Response 2: We are grateful for this critical and constructive comment. We completely agree that quantitative data are essential to robustly demonstrate the therapeutic potential of CSO. We apologize for the oversight in not including these key numerical results in our original manuscript. In direct response to this comment, we have now comprehensively revised the Results section of the Abstract (Line 29-36) to include the most significant numerical findings. Furthermore, to ensure consistency and provide full transparency, we have also enriched the main Results section (Section 2.2,Line 157 & 159) of the manuscript with corresponding detailed statistical data. The updated abstract now includes key quantitative findings, such as the time of treatment, the precise number of modulated gut microbes (4 butyrate-producing probiotics at the genus level and 2 pathogenic bacteria), and the scale of network pharmacology overlap (142 common targets between CSO and RA), with NLRP3 identified as the core effector. We believe these additions, highlighted in the revised manuscript, now provide a clear, quantitative, and compelling account of our findings, significantly strengthening the support for our conclusions.Thank you again for this constructive feedback, which has helped us improve the manuscript.We are grateful to the reviewer for this valuable comment again. |
||||||||||||||||||||||||||||||||||||||||||||||||||||||||||||||||||||||||||||||||||||||||||||||||||||||||||||||||||||||||||||||||||||||
|
Comments 3: Introduction: Please elaborate the role of short chain fatty acids in RA, using support available from the recent literature. |
||||||||||||||||||||||||||||||||||||||||||||||||||||||||||||||||||||||||||||||||||||||||||||||||||||||||||||||||||||||||||||||||||||||
|
Response 3: We are grateful to the reviewer for this excellent suggestion to elaborate on the role of short-chain fatty acids (SCFAs) in RA. We agree that a more detailed discussion of the role of SCFAs, particularly butyrate, in RA would provide a stronger rationale for our study. Accordingly, we have thoroughly revised the Introduction section (Line73-83) to provide a more detailed and mechanistic explanation. We now specifically discuss how SCFAs, particularly butyrate, modulate the Treg balance, inhibit cytokine production, and strengthen the gut barrier to improve inflammation. These additions, supported by recent key literature, firmly establish the scientific rationale for investigating SCFA-mediated mechanisms in our study on the therapeutic effects of CSO. |
||||||||||||||||||||||||||||||||||||||||||||||||||||||||||||||||||||||||||||||||||||||||||||||||||||||||||||||||||||||||||||||||||||||
|
Comments 4: Please also add what is in the oil that is responsible for therapeutic potential. |
||||||||||||||||||||||||||||||||||||||||||||||||||||||||||||||||||||||||||||||||||||||||||||||||||||||||||||||||||||||||||||||||||||||
|
Response 4: We thank the reviewer for this insightful comment. We agree that specifying the bioactive components responsible for the therapeutic effects of Coix Seed Oil (CSO) is crucial. In response, we have revised the relevant paragraph in the Introduction (Line 101-107) to detail the key phytochemicals in CSO. The updated text now highlights that CSO's efficacy is attributed to its high content of unsaturated fatty acids (e.g., oleic, linoleic acid and so on) and unique functional esters like coixenolide, with cited literature supporting their anti-inflammatory, microbiota-regulating and immunomodulatory activities. This addition provides a clear chemical basis for the therapeutic potential of CSO investigated in our study. |
||||||||||||||||||||||||||||||||||||||||||||||||||||||||||||||||||||||||||||||||||||||||||||||||||||||||||||||||||||||||||||||||||||||
|
Comments 5: Please determine the chemical composition of oil and correlated the active components with the claimed therapeutic effect. Without chemical characterization, it is not appropriate to report the therapeutic potential. |
||||||||||||||||||||||||||||||||||||||||||||||||||||||||||||||||||||||||||||||||||||||||||||||||||||||||||||||||||||||||||||||||||||||
|
Response 5: We deeply appreciated the reviewer for this critical comment, which we agree is fundamental to validating our study. In direct response, we have now included a comprehensive chemical characterization of the specific Coix Seed Oil (CSO) batch used in this study in the Result section (Section 2.1, line 122-151) and Materials and Methods section (Section 4.1&4.2, Line 550-583). As detailed therein, the composition was determined by UHPLC-OE/MS, which confirmed that our CSO is rich in unsaturated fatty acids, primarily oleic acid, linoleic acid, cholic acid and Indirubin. These quantified components are well-known for their anti-inflammatory, microbiota-modulatory and immunomodulatory activities, as cited in the revised Introduction (Line 101-107). Therefore, we have now directly correlated the characterized active components of our CSO with the investigated therapeutic effects against RA. |
||||||||||||||||||||||||||||||||||||||||||||||||||||||||||||||||||||||||||||||||||||||||||||||||||||||||||||||||||||||||||||||||||||||
|
Comments 6: In method LC-MS method is not appropriately described, please add the specifications of column, MS conditions and target range. Moreover, in results include the table showing composition and add in discussion the main components responsible for RA remedy. |
||||||||||||||||||||||||||||||||||||||||||||||||||||||||||||||||||||||||||||||||||||||||||||||||||||||||||||||||||||||||||||||||||||||
|
Response 6: We are thankful for these critical suggestions to improve the methodological rigor and chemical relevance of our work. We have thoroughly revised the manuscript to address all points as follows: a. LC-MS Method: The Materials and Methods section (Section 4.2, Preparation and UHPLC-MS analysis for CSO Extract; Section 4.12, Fecal metabolic analysis; Section 4.13, SCFAs quantification in serum) has been expanded to include the detailed specifications of the chromatographic column, mass spectrometric conditions, and the target mass range for analysis. b. Chemical Composition: Two new tables (Table 1-2) has been added to the Results section, summarizing the main components identified in CSO, including their identification data and relative abundances. c. Discussion on Main Components: The Discussion section (Line 394-400) has been revised to specifically correlate the key identified components with the observed anti-RA effects, citing relevant literature on their mechanisms of action. The revised content is stated as follows: (Section 4.2, Line 563-583) 4.2. Preparation and UHPLC-MS analysis for CSO Extract Prior to UHPLC-OE/MS analysis, protein precipitation was performed to extract metabolites.Briefly, after thawing, samples were centrifuged at 4°C and 12,000 rpm (RCF 13,800 ×g) for 15 min. A 300 μL aliquot of the supernatant was then mixed with 1000 μL of ice-cold extraction solvent (methanol:acetonitrile:water, 2:2:1, v/v/v) containing a mixture of isotope-labeled internal standards. The solution was vortexed vigorously for 30 s and subsequently sonicated in an ice-water bath for 5 min. To ensure complete protein precipitation, the mixture was incubated at -20°C for 1 h, followed by centrifugation again under the same conditions (4°C, 12,000 rpm for 15 min). The final supernatant was filtered through a 0.22 μm nylon membrane prior to UHPLC-MS/MS analysis. Chromatographic separation was achieved on a Vanquish UHPLC system (Thermo Fisher Scientific) using a Phenomenex Kinetex C18 column (2.1 mm × 100 mm, 2.6 μm) maintained at 4°C. The mobile phase consisted of (A) water with 0.01% acetic acid and (B) a mixture of isopropanol and acetonitrile (1:1, v/v). The injection volume was 2 μL. Mass spectrometric detection was carried out on an Orbitrap Exploris 120 instrument (Thermo Fisher Scientific) controlled by Xcalibur software (version 4.4). The instrument was operated in data-dependent acquisition (DDA) mode. Key MS parameters were set as follows: sheath gas flow rate, 50 Arb; auxiliary gas flow rate, 15 Arb; capillary temperature, 320 °C; full MS resolution, 60,000; MS/MS resolution, 15,000; stepped normalized collision energy (SNCE), 20/30/40; spray voltage, +3.8 kV (positive ion mode) or -3.4 kV (negative ion mode). (Section 4.12, Line 655-669) Frozen intestinal content (~60 mg) was homogenized with steel beads in 600 μL of cold methanol-water (4:1, v/v) containing 20 μL of internal standard (L-2-chlorophenylalanine, 0.3 mg/mL) after 5 min at -20°C. The mixture was ultrasonicated (ice-water bath, 10 min), incubated at -20°C for 30 min, and centrifuged (13,000 rpm, 4°C, 10 min). A 300 μL aliquot of supernatant was dried, reconstituted in 400 μL methanol-water (1:4, v/v), vortexed, sonicated, and incubated at -20°C for 2 h. After a final centrifugation, 150 μL of supernatant was filtered (0.22 μm) into an LC vial and stored at -80°C until analysis. A pooled QC sample was prepared from all extracts. All solvents were pre-chilled to -20°C. LC-MS analysis was performed using an AB ExionLC UPLC system coupled to a Q Exactive mass spectrometer. Separation used an ACQUITY UPLC HSS T3 column (100 × 2.1 mm, 1.8 μm) at 45°C, with mobile phase A (water with 0.1% formic acid) and B (acetonitrile with 0.1% formic acid) at 0.35 mL/min. Injection volume was 2 μL. ESI source parameters were: spray voltages (positive/negative ion modes), sheath and auxiliary gas settings, and capillary temperature as per instrument method. Data were acquired in full scan mode. (Section 4.13, Line 672-684) The SCFAs in plasma samples, including acetic acid, butyric acid, isovaleric acid and pentanoic acid were comprehensively analyzed. Target metabolites were qualitatively and quantitatively analyzed using a UPLC-ESI-MS/MS system. Chromatographic separation was performed on an Acquity UPLC BEH C18 column (2.1 × 100 mm, 1.7 µm) maintained at 40°C. The mobile phase consisted of 0.1% formic acid in water (solvent A) and acetonitrile (solvent B), delivered at a flow rate of 0.35 mL/min with an injection volume of 5 µL. The gradient elution program was as follows: 0-1 min, 90% A; 1-2 min, 90% to 75% A; 2-6 min, 75% to 65% A; 6-6.5 min, 65% to 5% A; 6.5-7.8 min, hold at 5% A; 7.8-7.81 min, 5% to 90% A; 7.81-8.5 min, hold at 90% A. Mass spectrometric detection was conducted using an electrospray ionization (ESI) source. The ion source temperature was set at 450°C, with spray voltages of +5500 V and -4500 V for positive and negative ion modes, respectively. The curtain gas pressure was 35 psi, the nebulizing gas (Gas1) pressure was 50 psi, and the auxiliary heating gas (Gas2) pressure was 60 psi. The collision-activated dissociation (CAD) parameter was set to "Medium". (Section 2.1, Line 132-137) Table 1: Positive ingredients of CSO identified by UHPLC-OE/MS
(Section 2.1, Line 139-150) Table 2: Negative ingredients of CSO identified by UHPLC-OE/MS
|
||||||||||||||||||||||||||||||||||||||||||||||||||||||||||||||||||||||||||||||||||||||||||||||||||||||||||||||||||||||||||||||||||||||
|
Comments 7: The result and discussion sections are appropriate and well presented. |
||||||||||||||||||||||||||||||||||||||||||||||||||||||||||||||||||||||||||||||||||||||||||||||||||||||||||||||||||||||||||||||||||||||
|
Response 7: We are grateful to the reviewer for the kind and positive comments regarding the clarity and appropriateness of our results and discussion. We are pleased that the core findings of our study are presented clearly, and we hope these results will contribute meaningfully to the future research in this field. |
||||||||||||||||||||||||||||||||||||||||||||||||||||||||||||||||||||||||||||||||||||||||||||||||||||||||||||||||||||||||||||||||||||||
|
Comments 8: Overall the manuscript contains significant contribution, however above-mentioned points should be addressed. |
||||||||||||||||||||||||||||||||||||||||||||||||||||||||||||||||||||||||||||||||||||||||||||||||||||||||||||||||||||||||||||||||||||||
|
Response 8: We would like to thank the reviewer once more for the constructive comments, which have been instrumental in strengthening the presentation of our work. We are also grateful for your time and effort, and for your encouraging statement regarding the significant contribution of our manuscript. We have incorporated all the suggestions and corrections as detailed in the point-by-point responses above. We believe that the revisions have significantly improved the quality of our manuscript and hope the manuscript now meets your expectations. |
||||||||||||||||||||||||||||||||||||||||||||||||||||||||||||||||||||||||||||||||||||||||||||||||||||||||||||||||||||||||||||||||||||||

Reviewer 3 Report
Comments and Suggestions for Authors
This paper should be resubmitted as two separate papers: one focusing on experiments conducted with rats, and another with mice. Furthermore, the paper should be resubmitted as two separate papers. The first paper should demonstrate that coix seed oil (COS) inhibits the development of arthritis (AR) through the suppression of the production of inflammatory cytokines. The second paper should explicitly show that the AR-suppressing effect of COS is closely related to changes in butyrate metabolism induced by COS. I will review the resubmitted papers to determine their suitability for publication in the journal. Therefore, I do not provide detailed evaluations of this paper.
Author Response
Please see the attachment.
|
Response to Reviewer 3 Comments
|
||
|
1. Summary |
|
|
|
Thank you very much for your careful review of our manuscript. In response, we have revised the manuscript with careful consideration to enhance its accessibility and impact. Below, we provide detailed responses. We hope that the revised manuscript and our responses adequately address the issues you raised and meet your expectations. |
||
|
2. Questions for General Evaluation |
Reviewer’s Evaluation |
Response and Revisions |
|
Does the introduction provide sufficient background and include all relevant references? |
Must be improved |
Please see as described below. |
|
Is the research design appropriate? |
Must be improved |
Please see as described below. |
|
Are the methods adequately described? |
Must be improved |
Please see as described below. |
|
Are the results clearly presented? |
Must be improved |
Please see as described below. |
|
Are the conclusions supported by the results? |
Must be improved |
Please see as described below. |
|
Are all figures and tables clear and well-presented? |
Must be improved |
Please see as described below. |
|
3. Point-by-point response to Comments and Suggestions for Authors |
||
|
Comment : This paper should be resubmitted as two separate papers: one focusing on experiments conducted with rats, and another with mice. Furthermore, the paper should be resubmitted as two separate papers. The first paper should demonstrate that coix seed oil (COS) inhibits the development of arthritis (AR) through the suppression of the production of inflammatory cytokines. The second paper should explicitly show that the AR-suppressing effect of COS is closely related to changes in butyrate metabolism induced by COS. I will review the resubmitted papers to determine their suitability for publication in the journal. Therefore, I do not provide detailed evaluations of this paper. |
||
|
Response: We profoundly grateful to the reviewer for this rigorous and thought-provoking comment, as well as your highly constructive and insightful suggestions regarding the logical framework of our study. Regarding your recommendation to split the manuscript into two independent studies: one on "direct anti-inflammatory effects" and "gut microbiota-metabolite mechanisms," and another on "rats" and "mice", we have given it careful and thorough consideration and discussion. We recognize that the intention is to enhance the clarity and rigor of the scientific findings, and we sincerely agree and are deeply grateful for this suggestion. After carefully considering your suggestions, we recognize that our article has shortcomings in presenting our research approach, and our data does not clearly demonstrate the inherent logic we initially designed. We sincerely apologize for any confusion caused. We would like to further explain why we believe that integrating the two parts of the study into a single paper is the best way to convey a complete, coherent, and more causally persuasive scientific narrative. We are not rejecting your professional opinion; rather, we aim to fully address your core concerns while preserving the integrity of the manuscript through additional literature and refinement of the paper. We respond primarily through the following points: 1. Regarding Research Approach and Study Design Issues This study, through a systematic and rigorous experimental design, progressively reveals the key core aspects of the research. We aim to demonstrate that coix seed oil (CSO) alleviates the progression of rheumatoid arthritis (RA) by inhibiting the release of NLRP3, a mechanism closely related to CSO's modulation of the gut microbiota and butyrate levels. The NLRP3 inflammasome plays a central role in the inflammatory response and bone destruction in rheumatoid arthritis [1]. In our study, we found that CSO exerts a certain inhibitory effect on NLRP3. Furthermore, differences in the gut microbiota aggravate inflammation in CIA mice. We aim to illustrate that variations in the gut microbiota contribute to the development of joint inflammation and also have a certain effect on NLRP3, which is closely associated with joint inflammation. Both approaches attempt to elucidate the same phenomenon and content; however, it is evident that we have not explained this adequately. Therefore, we have provided additional clarification to better illustrate the relationship between the two. We have also supplemented with a correlation analysis figure (Figure 9K) to clearly show that the differential microbiota regulated by CSO is closely associated with NLRP3. A schematic diagram (Graphical Abstract) intuitively summarizes how the modulation of the gut microbiota/short-chain fatty acids and the inhibition of the NLRP3 inflammasome converge to improve arthritis. We sincerely appreciate your keen observation in identifying our shortcomings, which has guided the direction for further research. We will further explore the causal relationship between the two in the future. 2. Regarding the Selection of Rats and Mice for the Experiment Both the SD rat and DBA mouse models of collagen-induced arthritis (CIA) are gold standard preclinical models and have a high degree of pathophysiological similarity to human RA. They replicate key disease features, including joint swelling, synovial hyperplasia, inflammatory cell infiltration, and cartilage and bone destruction. [2] Rats possess unique advantages in pharmacodynamic studies of arthritis due to their larger body size, which results in bigger joints and more substantial synovial tissue. Consequently, the arthritis index (AI) and pathological changes are more clearly observable in rats than in mice, facilitating the assessment of CSO effects on inflammation, pathological alterations, and bone destruction (joint CT) in the CIA model. Therefore, we selected the rat model for pharmacodynamic investigations. Furthermore, rats allow for the collection of greater volumes of blood and feces, enabling serum inflammatory factor assays and serum-targeted metabolomics, as well as fecal untargeted metabolomics and fecal 16S rRNA analysis, providing corresponding data for this study. Accordingly, SD rats were chosen for CIA model induction and efficacy evaluation in this research. We strategically transitioned to a mouse model to conduct corresponding mechanism studies. First, the DBA mouse strain is also highly sensitive to the CIA collagen-induced model, and the modeling rate is high. In addition, in our previous studies, it has been clearly confirmed that CSO also has a therapeutic effect on DBA-induced model mice [3]. Compared with rats, mice produce less butyrate from their own intestinal flora, but the absorption, distribution, metabolism, and mechanism of action of butyrate in mice are more consistent with those of humans, and the research results can be better extrapolated to humans. [4] In order to further clarify the effect of butyrate on RA, we chose mice when we gave exogenous butyrate, and the goal was to establish that butyrate supplementation can effectively improve the core phenomenon of arthritis. On the basis of proving the effectiveness of butyrate, we will use gene knockout operation, a sterile mouse model, and other mouse genetic mechanism characteristics to deeply explore the specific mechanism of butyrate through which receptor, which type of immune cells, and how to regulate the intestinal barrier. Therefore, the current mouse phenotype experiment provides an indispensable premise and foundation for the subsequent in-depth mechanism research. Finally, we sincerely thank the reviewers for raising the question regarding splitting the manuscript into two separate papers. We also cordially invite the reviewers to re-evaluate our manuscript. This paper systematically presents the comprehensive mechanisms of CSO in treating RA, including the improvement of the gut microbiota/butyrate axis and the direct inhibition of the NLRP3 pathway to alleviate RA. In future studies, we will further investigate the potential interrelated effects and underlying mechanisms, providing a more complete research framework. We believe that the unified manuscript in its current form offers a more influential, coherent, and compelling narrative. We are deeply grateful for the opportunity to revise and improve our work; your feedback has been invaluable and has greatly helped us enhance the logical presentation of our study. We respectfully request that the reviewers consider our explanation based on scientific rationale and grant us the opportunity to revise the entire manuscript according to this framework and resubmit it. |
||

Reviewer 4 Report
Comments and Suggestions for Authors
The novelty of the study lies in the detailed elucidation of the mechanism by which Coix seed oil attenuates rheumatoid arthritis, namely through the identification of a previously unrecognized gut–butyrate–joint axis and the involvement of the NLRP3 inflammasome pathway. The beneficial effects of Coix seed oil are mediated by an increase in butyrate production by the gut microbiota, inhibition of NLRP3-dependent inflammatory signaling, and a consequent reduction in bone destruction.
My literature survey indicates that the work is innovative in this respect. The authors employed an integrated multi-omics strategy to map the mechanism of action of Coix seed oil, which is a strength of the study. They used a broad panel of complementary analytical methods that are well chosen and appropriate for the research questions posed. In my opinion, the manuscript is scientifically valuable. I did not identify any major methodological or interpretative flaws. The topic is relevant in the field. The manuscript is clearly written, including the description of the experimental procedures. The methodology, presentation of results, discussion, and reference list are appropriate and adequately support the authors’ conclusions.
Some drawings are too small. In the References section, sometimes only a single page number is given, whereas in other cases a page range is provided.
It is a very nice paper.
Author Response
Please see the attachment.
|
Response to Reviewer 4 Comments
|
||
|
1. Summary |
|
|
|
We are grateful for the time and effort you invested in reviewing our manuscript and your insightful comments. In response, we have revised the manuscript with careful consideration to enhance its clarity and impact. Below, please find our point-by-point detailed responses (in purple), with all corresponding changes highlighted in red within the revised manuscript. We are hopeful that these revisions have adequately all the issues you raised. |
||
|
2. Questions for General Evaluation |
Reviewer’s Evaluation |
Response and Revisions |
|
Does the introduction provide sufficient background and include all relevant references? |
Yes |
Please see as described below. |
|
Is the research design appropriate? |
Yes |
Please see as described below. |
|
Are the methods adequately described? |
Yes |
Please see as described below. |
|
Are the results clearly presented? |
Yes |
Please see as described below. |
|
Are the conclusions supported by the results? |
Yes |
Please see as described below. |
|
Are all figures and tables clear and well-presented? |
Can be improved |
Please see as described below. |
|
3. Point-by-point response to Comments and Suggestions for Authors |
||
|
Comments 1: The novelty of the study lies in the detailed elucidation of the mechanism by which Coix seed oil attenuates rheumatoid arthritis, namely through the identification of a previously unrecognized gut–butyrate–joint axis and the involvement of the NLRP3 inflammasome pathway. The beneficial effects of Coix seed oil are mediated by an increase in butyrate production by the gut microbiota, inhibition of NLRP3-dependent inflammatory signaling, and a consequent reduction in bone destruction. |
||
|
Response 1: We thank the reviewer for their positive and insightful comments on our work. We are particularly grateful that they have accurately identified and articulated the core novelty of our study—the elucidation of the gut–butyrate–joint axis and the involvement of the NLRP3 inflammasome pathway as a key mechanism through which Coix seed oil exerts its therapeutic effects against rheumatoid arthritis. |
||
|
Comments 2: My literature survey indicates that the work is innovative in this respect. The authors employed an integrated multi-omics strategy to map the mechanism of action of Coix seed oil, which is a strength of the study. They used a broad panel of complementary analytical methods that are well chosen and appropriate for the research questions posed. In my opinion, the manuscript is scientifically valuable. I did not identify any major methodological or interpretative flaws. The topic is relevant in the field. The manuscript is clearly written, including the description of the experimental procedures. The methodology, presentation of results, discussion, and reference list are appropriate and adequately support the authors’ conclusions. |
||
|
Response 2: We are deeply grateful to the reviewer for their exceptionally positive and encouraging evaluation of our work. The recognition of the innovative nature of our findings, the strength of our integrated multi-omics strategy, and the overall scientific value and clarity of the manuscript is immensely encouraging to us. We sincerely thank the reviewer for their time and for this thorough and supportive assessment. |
||
|
Comments 3: Some drawings are too small. In the References section, sometimes only a single page number is given, whereas in other cases a page range is provided. |
||
|
Response 3: We thank the reviewer for their positive overall assessment and for these helpful technical suggestions. a. Regarding the figure sizes, we have carefully reviewed all drawings and increased the size of the ones that were too small to ensure all details are clearly visible (Figure1, Figure4, Figure6, and Figure9). b. Regarding the references, we have thoroughly checked and unified the format of the entire reference list, ensuring that all entries now provide the complete page ranges. We believe these corrections have further improved the clarity and professionalism of the manuscript. |
||
|
Comments 4: It is a very nice paper. |
||
|
Response 4: We are delighted and honored to receive such positive feedback from the reviewer. We sincerely thank the reviewer for the time and for the encouraging comments, which are a great motivation for our team. We hope that the revised manuscript now fully address your request. Thank you for considering our work. |
||

Round 2
Reviewer 2 Report
Comments and Suggestions for Authors
The manuscript has been comprehensively revised as advised, all the components including chemical characterization of Oil are now included and study now qualifies for inclusion as a publication.
Reviewer 3 Report
Comments and Suggestions for Authors
Upon reviewing the revised version of manuscript, it is clear that the authors have completely addressed to my comments and revised the entire manuscript. However, the revised version of the manuscript remains insufficient for publication in the journal.
Major comments:
As stated in your response to reviewer’s comments, if the authors do not wish to split this manuscript into two papers—one using rats and the other focusing on mice—please revise the manuscript according to the following points.
1) You should speculate the suppressive mechanisms of CSO on the activation of NLRP3 observed in experimental mice.
2) You should discuss the relationship between the improvement of the gut microbiota/butyrate axis and inhibition of NLRP3 activation.
Minor points:
1) The concentration of CSO used for treatment differed between rats and mice. Why was this concentration different? Please explain. Alternatively, please explain the rationale for determining the concentration of the drug used for treatment.
2) Please clarify the methods for sample preparation used for RT-PCR and PCR conditions.
Round 3
Reviewer 3 Report
Comments and Suggestions for Authors
I have confirmed that the authors have thoroughly revised the manuscript in accordance with my comments.
I am grateful for the opportunity to review such an interesting manuscript.